

# Satellite products of incoming solar and longwave radiations used for snowpack modelling in mountainous terrain

Louis Quéno[1], Fatima Karbou[1], Vincent Vionnet[1], and
Ingrid Dombrowski-Etchevers[2]

[1]Météo-France/CNRS, CNRM UMR3589, CEN, St. Martin d'Hères, France
[2]Météo-France/CNRS, CNRM UMR3589, Toulouse, France

*Correspondence to:* Louis Quéno (louis.queno@meteo.fr)

**Abstract.** In mountainous terrain, the snowpack is strongly affected by incoming shortwave and longwave radiations. In this study, a thorough evaluation of the incoming solar and longwave radiation products (DSSF and DSLF) derived from the Meteosat Second Generation satellite was undertaken in the French Alps and the Pyrenees. The satellite products were compared with forecast

fields from the meteorological model AROME and with analyses fields from the SAFRAN system. A new satellite-derived product (DSLFnew) was developed by combining satellite observations and AROME forecasts. An evaluation against in situ measurements showed lower errors for DSSF than AROME and SAFRAN in terms of solar irradiances. For longwave irradiances, contrasted results falling in the range of uncertainty of sensors did not enable us to select the best product. Spatial

comparisons of the different datasets over the Alpine and Pyrenean domains highlighted a better representation of the spatial variability of solar fluxes by DSSF and AROME than SAFRAN. We also showed that the altitudinal gradient of longwave irradiance is too strong for DSLFnew and too weak for SAFRAN. These datasets were then used as radiative forcing together with AROME near-surface forecasts to drive distributed snowpack simulations by the model Crocus in the French

Alps and the Pyrenees. An evaluation against in-situ snow depth measurements showed higher biases when using satellite-derived products, despite their quality. This effect is attributed to some error compensations in the atmospheric forcing and the snowpack model. However, satellite-derived radiation products are judged beneficial for snowpack modelling in mountains, when the error compensations are solved.

# 1 Introduction

Seasonal snowpacks are a key component of mountain hydrological systems. Snow accumulation and ablation processes set up the temporal evolution of the snow cover and its spatial distribution, controlling the snow melt variability and timing, which govern the run-off in high-altitude catch-



ments (e.g. Anderton et al., 2002; DeBeer and Pomeroy, 2017). The evolution and spatial distribu-
tion of the snowpack in mountainous terrain depends on its energy budget, affected by the surface
radiative budget, the sensible and latent heat fluxes and the ground heat flux (e.g. Armstrong and
Brun, 2008). The meteorological conditions are the main factors controlling the snow surface energy
budget, with a key contribution of the radiative components (Male and Granger, 1981). For exam-
ple, Cline (1997) reported a contribution of 75% of net radiative fluxes in the energy for snowmelt
over the entire season at a continental midlatitude alpine site of the Colorado Front Range (3517
m), while Marks and Dozier (1992) found a contribution between 66% and 90% at two alpine sites
of the Sierra Nevada (2800 m and 3416 m). Therefore, incoming shortwave (SW↓) and longwave
(LW↓) radiations are amongst the most significant atmospheric factors of the energy and mass bud-
get of the snowpack, particularly during snowmelt periods. It is crucial to accurately represent them
in numerical snowpack simulations, as recent works underlined the strong sensitivity of snowpack
simulations to the radiative forcing (Raleigh et al., 2015; Lapo et al., 2015b; Sauter and Obleitner,
2015).

Several studies highlighted the benefits of distributed snowpack simulations at the scale of moun-
tain ranges, particularly in areas with scarce snow cover observations. Simulations of detailed snow-
pack models driven by Numerical Weather Prediction (NWP) forecasts at kilometric resolution were
proved to describe satisfactorily the snowpack variability within a mountain range (Quéno et al.,
2016; Vionnet et al., 2016), the snow accumulation quantitative distribution (Schirmer and Jamieson,
2015), and to provide relevant high-resolution information for snowpack stability concerns (Bellaire
et al., 2014; Horton et al., 2015). The radiative forcing of these simulations relies on NWP forecasts
of the SW↓ and LW↓ radiations with no use of observations (in situ or from satellites). Vionnet et al.
(2016) made a preliminary evaluation of SW↓ and LW↓ radiation forecasts by the NWP system
AROME operating at 2.5 km resolution over France. Through comparisons to ground-based mea-
surements at two mountainous sites in the French Alps, they showed an overestimation of SW↓ and
an underestimation of LW↓, linked to an underestimation of the cloud cover.

Satellite-derived estimates of SW↓ and LW↓ irradiances are an alternative to NWP-based radiation
datasets in mountainous terrain. They are mostly based on satellite products of cloud mask, which
highly controls the incoming radiations in mountains (e.g. Sicart et al., 2016), and top-of-atmosphere
reflectances. These satellite-based products could have a potential added value for snowpack mod-
elling since they are available continuously and at a relatively high resolution in mountains, where in
situ observations are rather scarce. This approach has already been explored with the solar and long-
wave surface irradiance data from NASA's Clouds and the Earth's Radiant Energy System synoptic
(CERES SYN; Rutan et al., 2015), which are satellite-derived estimates at 3 h temporal resolution
and 1°grid spacing (i.e. approximately 110 km at midlatitudes). The scores of CERES SYN irra-
diances were found to be poorer at mountain stations than in plains (Hinkelman et al., 2015). The
CERES SYN solar irradiance product was also evaluated by Lapo et al. (2017) who found large



biases over complex terrain. Hinkelman et al. (2015) used CERES SYN irradiance products to drive snowmelt simulations in complex terrain and found performances in the range of empirical methods and observations. In this study, we used the SW↓ and LW↓ irradiances from the Satellite Application Facility on Land Surface Analysis (LSA SAF; Trigo et al., 2011). These products have a higher tem-
poral frequency (30 min) and a higher spatial resolution (3 km), and thus may be more adapted than CERES SYN products to complex terrains, where the subgrid variability of incoming radiations within a 1°grid cell is the highest (Hakuba et al., 2013). In a perspective of distributed snowpack simulations at kilometric resolution, they are also consistent with the horizontal resolution of the other atmospheric variables from NWP systems. LSA SAF irradiance products were proved to be
valuable in plains (e.g. Geiger et al., 2008b; Ineichen et al., 2009; Trigo et al., 2010; Carrer et al., 2012; Moreno et al., 2013; Cristóbal and Anderson, 2013), with a significant positive impact when used for soil simulations (Carrer et al., 2012) or evapotranspiration modelling (Ghilain et al., 2011; Sun et al., 2011).

The aim of the present study is to assess LSA SAF products of SW↓ and LW↓ radiations in the
French Alps and the Pyrenees, and to compare them with kilometric-resolution NWP forecasts and with a meteorological analysis system dedicated to mountainous terrain. We also test and discuss the potential of LSA SAF irradiance products to drive distributed snowpack simulations in mountains.

## 2 Data and models

### 2.1 Study domain and period

The study focuses on two domains covering the French Alps (Fig. 1a) and the French and Spanish Pyrenees (Fig. 1b). The French Alps domain ranges from 43.125°N to 46.875°N latitudes and from 4.5°E to 8.5°E longitudes. This domain also includes a part of the mid-altitude mountain range of Jura. The Pyrenees domain covers the latitudes from 41.6°N to 43.6°N and the longitudes from - 2.5°E to 3.5°E. Hourly data, from 1 August 2010 to 31 July 2014, including in situ measurements,
satellite irradiance products, meteorological models and snowpack simulations were used.

### 2.2 Irradiance datasets

Several irradiance datasets were used in this study: forecasts from the NWP model AROME, reanalyses from the SAFRAN analysis system, LSA SAF irradiance products derived from remotely-sensed observations and a hybrid LW↓ irradiance product based on a combination of LSA SAF algorithms
with AROME forecasts. An in situ observation dataset was built up for validation in mountains.

### 2.2.1 NWP system: AROME

AROME (Application of Research to Operations at MEsoscale) is the meso-scale NWP system of Météo-France (Seity et al., 2011), operating over France since December 2008 at 2.5 km grid



spacing (1.3 km since 2015; Brousseau et al., 2016). It is a spectral and non-hydrostatic model.
The physics and data assimilation schemes are detailed in Seity et al. (2011). In particular, AROME
uses the radiation parametrizations from the European Centre for Medium-Range Weather Forecasts
(ECMWF), with the SW scheme from Fouquart and Bonnel (1980) and the LW scheme from Mlawer
et al. (1997).

In this study, we built a continuous atmospheric forcing dataset using hourly AROME forecasts
issued from the 0 UTC analysis time, from + 6 h to + 29 h, extracted on a regular latitude/longitude
grid with a 0.025°resolution over the period and domains of study (Sect. 2.1, Fig. 1), similarly to
Quéno et al. (2016) and Vionnet et al. (2016). Besides incoming shortwave and longwave radiations,
2 m temperature and humidity, as well as 10 m wind speed and precipitation (amount of rainfall and
snowfall) are part of the AROME forcing.

**2.2.2   Analysis system: SAFRAN**

SAFRAN (Système d'Analyse Fournissant des Renseignements Atmosphériques à la Neige; Anal-
ysis System Providing Atmospheric Information to Snow; Durand et al., 1993, 2009a, b) is a me-
teorological analysis system developed to provide hourly estimation of meteorological parameters
required to drive land surface models. SAFRAN outputs are available per 300 m altitude steps within
mountainous regions called "massifs". There are 23 massifs in the French Alps and 23 massifs in
the French and Spanish Pyrenees (Fig. 1), defined for their climatological homogeneity. SAFRAN
reanalyses take a preliminary guess from the global NWP model ARPEGE (from Météo-France, 15
km grid spacing projected on a 40 km grid; Courtier et al., 1991) combined by optimal interpolation
with available observations from automatic weather stations, manual observations carried out in the
climatological network and in ski resorts, remotely-sensed cloudiness and atmospheric upper-level
soundings. In particular, the incoming shortwave and longwave radiations are computed with the
radiation scheme from Ritter and Geleyn (1992), using as first guess vertical profiles of temperature
and humidity from ARPEGE forecasts, atmospheric soundings, a guess of cloudiness based on the
analysed vertical humidity profile and a cloud mask detected by satellite (Derrien et al., 1993).

In this study, we used SAFRAN reanalyses from 1 August 2010 to 31 July 2014. For compar-
isons to in situ irradiance observations, the reanalyses were interpolated at the exact elevation of the
stations. For La Pesse station in Jura (Fig. 1a), the extension of SAFRAN to mid-altitude French
massifs (Lafaysse et al., 2013) was used. For Carpentras station in plains (Fig. 1a), the SAFRAN-
France extension (Quintana-Seguí et al., 2008) was considered. For distributed comparisons and
for the atmospheric forcing of distributed snowpack simulations, the reanalyses at massif-scale in
the French Alps and in the Pyrenees were interpolated over the 0.025°grid of the AROME forcing,
within SAFRAN massifs, similarly to Quéno et al. (2016) and Vionnet et al. (2016), following the
method described in Vionnet et al. (2012).



### 2.2.3 LSA SAF products

The LSA SAF is a project supported by the European Organisation for the Exploitation of Meteorological Satellites (EUMETSAT) and a consortium of European National Meteorological Services, with the purpose to use remotely-sensed data to determine land surface variables (Trigo et al., 2011). In particular, it provides estimates of the Downward Surface Shortwave Flux (DSSF) and the Downward Surface Longwave Flux (DSLF), derived from the Spinning Enhanced Visible and

Infrared Imager (SEVIRI) radiometer on board the Meteosat Second Generation (MSG) geostationary satellite (Schmetz et al., 2002). They are generated every 30 min, covering the MSG full disk with a 3 km resolution at nadir. They have been operationally disseminated since September 2005 (http://landsaf.ipma.pt). DSSF and DSLF are fully consistent as they are based on the same satellite observations.

• SW↓ irradiance: DSSF

The algorithm to estimate the DSSF is described in details by Geiger et al. (2008b). The MSG/SEVIRI cloud mask (Derrien and Le Gléau, 2005) identifies clear-sky and cloudy-sky situations. Two separate algorithms are then applied. In the clear-sky method, derived from Frouin et al. (1989), the effective transmittance of the atmosphere is computed using the total

column water vapour content (TCWV) forecast by the European Centre for Medium-Range Weather Forecasts (ECMWF) Integrated Forecasting System (IFS), the ozone amount from the Total Ozone Mapping Spectrometer climatology, a constant visibility and the surface albedo taken from the LSA SAF albedo product (Geiger et al., 2008a). In the cloudy-sky method, derived from Gautier et al. (1980) and Brisson et al. (1999), the top-of-atmosphere reflectance

observed by MSG/SEVIRI is used in addition to the former set of variables. The target accuracy of the DSSF is 10% or 20 W m$^{-2}$ for values lower than 200 W m$^{-2}$.

• LW↓ irradiance: DSLF

The algorithm to estimate the DSLF is described in details by Trigo et al. (2010). It consists in a modified version of the bulk parametrization of Prata (1996), initially developed for clear skies

only. It relies on a formulation of the effective emissivity and temperature of the atmospheric layer above the surface, using the TCWV, 2 m temperature ($T_{2m}$) and 2 m dew point ($Td_{2m}$) forecast by the ECMWF IFS. The formulation parameters are calibrated for clear-sky and overcast conditions independently. The MSG/SEVIRI cloud mask (Derrien and Le Gléau, 2005) is thus the only observation used, to distinguish clear-sky and cloudy-sky situations. In

case of partly cloudy situations, the average of both terms is taken. The target accuracy of the DSLF is 10%.



### 2.2.4 New DSLF product using AROME forecasts

The DSLF relies on the ECMWF IFS forecasts of TCWV, $T_{2m}$ and $Td_{2m}$. These atmospheric variables have a strong dependence on altitude and a strong spatial variability in mountainous terrain.

The 16-km horizontal resolution of the ECMWF IFS hardly represents this spatial variability in the Alps and the Pyrenees, despite a constant lapse rate applied for grid elevation correction. Consequently, we developed a new DSLF product using the same algorithm (Trigo et al., 2010) depending on the cloud mask (Derrien and Le Gléau, 2005), but replacing ECMWF forecasts by AROME forecasts at 2.5 km resolution, which provides a finer representation of the topography. Air temperature

and dew point were taken at the first operational atmospheric level. The use of AROME also implies a better agreement of the atmospheric forecast resolution (2.5 km) with the cloud mask and final product resolution (3 km).

AROME forecasts were interpolated over the LSA SAF grid through a closest-neighbour method (similar grid spacings). The possible altitude difference between AROME grid points and LSA SAF

grid points was mitigated thanks to a vertical temperature gradient of $- 6.5$ K km$^{-1}$, similarly to the method applied to ECMWF IFS forecasts. The algorithm was applied to the new DSLF on the LSA SAF grid over the domains of study (Fig. 1), from 1 August 2010 to 31 July 2014. Hereafter, this product is referred to as DSLFnew.

### 2.2.5 In situ irradiance observations

To assess the distributed irradiance datasets, ground measurements of SW↓ and LW↓ were extracted from Météo-France station network and additional Automatic Weather Stations (AWS). Stations with altitude higher than 1000 m were selected. Since the elevation is one of the most significant factor of surface radiation spatial variability (Oliphant et al., 2003), stations were not used for evaluation if the difference between the station elevation and the elevation of the four closest AROME and

LSA SAF grid points was higher than 300 m. The resulting observation database, represented in Fig. 1, includes 14 mountain SW↓ stations (8 in the French Alps, 1 in Jura and 5 in the Pyrenees), 4 mountain LW↓ stations (3 in the French Alps and 1 in the Pyrenees). An additional station located in plains at Carpentras (Fig. 1) has been included in the database since it is the reference station for SW↓ and LW↓ measurements in France. These stations and their characteristics are listed in Table 1.

Radiation measurements are scarce in mountainous terrain and their quality is often lower than plain measurements, due to the difficulty to maintain these stations and the possible occurrence of frost or snow on the sensors in winter (Lapo et al., 2015a). The pyranometers from Météo-France network (Kipp&Zonen CM5, CM6B and CM11) meet the good quality standards of the World Meteorological Organization (WMO, 2014), hence an uncertainty of hourly total SW↓ irradiance from

$\pm 5\%$ to $\pm 8\%$. Due to their location in altitude implying difficulties of maintenance, it seems more realistic to retain here a maximum uncertainty of $\pm 10\%$. The station of Carpentras in plains is equipped





with the pyranometer Kipp&Zonen CM21 and the pyrgeometer Kipp&Zonen CG4. This station is a reference station for radiation measurements, as it is part of the Baseline Surface Radiation Network (BSRN; Ohmura et al., 1998): the uncertainties are $\pm$3% for SW↓ and $\pm$5% for LW↓. At Col de

Porte where the pyranometer Kipp&Zonen CM14 and the pyrgeometer Kipp&Zonen CG4 undergo a regular maintenance, Morin et al. (2012) reported a total uncertainty on the order of $\pm$10% (including site-dependent uncertainties). The AWS of Bassiès (Szczypta et al., 2015), Argentière glacier and St-Sorlin glacier (data from GLACIOCLIM program, https://glacioclim.osug.fr) have Kipp&Zonen CM3 pyranometers and CG3 pyrgeometers, classified as moderate quality after WMO's standards

(WMO, 2014), for which the manufacturer reports a daily total accuracy of $\pm$10%. The uncertainties have not been estimated at these stations. They are possibly higher than 10% because of the difficulty to maintain AWS in complex environment, particularly in winter. WMO (2014) indicates uncertainties up to $\pm$20% for hourly totals for this kind of instruments. The results at these stations are indicative for high altitudes but shall be considered carefully. Table 1 summarizes the measure-

ment uncertainties at each station.

### 2.3 Snowpack datasets

The impact of the different irradiance datasets on distributed snowpack simulations is assessed using the snowpack model Crocus with different atmospheric forcings. These simulations are compared to in situ measurements of snow depth (SD).

#### 2.3.1 Snowpack model: Crocus

Snowpack simulations driven by different irradiance datasets were performed with the detailed snow cover model Crocus (Brun et al., 1992; Vionnet et al., 2012) coupled with the ISBA land surface model within the SURFEX simulation platform (Masson et al., 2013), to fully simulate the interactions between snowpack and soil. SURFEX/ISBA-Crocus (called Crocus hereafter) simulates the

evolution of the snowpack physical properties along its stratigraphy, under given atmospheric forcing data (temperature and specific humidity at a given height above the surface, wind speed at a given height above the surface, SW↓ and LW↓ irradiance, solid and liquid precipitation).

The simulations were carried out over the French Alps and Pyrenees domains (Fig. 1), on the AROME regular latitude/longitude grid at 0.025°resolution (Sect. 2.2.1) from 1 August 2010 to 31

July 2014. The effect of aspect and slope on incoming solar radiations were not taken into account, and the interactions with the vegetation and the parametrization of fractional snow cover were not activated, because the evaluation observations are supposed to be in flat and open fields. This configuration has already been used in Vionnet et al. (2016) and Quéno et al. (2016).

Except incoming radiations, the atmospheric forcing of the snowpack simulations was built with

AROME forecasts (Sect. 2.2.1). The radiative components of the forcings were extracted from the different irradiance datasets: a) AROME radiation forecasts (simulations named A-Cro hereafter),



b) SAFRAN radiation reanalyses (simulations named AS-Cro hereafter), c) DSSF and DSLFnew (simulations named AL-Cro hereafter). In order to include DSSF and DSLFnew products in AROME forcing, the interpolation on AROME grid was made to minimize the effect of elevation difference on the incoming radiations. Among the four nearest LSA SAF grid points, the grid point with the minimum altitude difference with AROME grid point was chosen. Similarly to Hinkelman et al. (2015), SW↓ irradiances were not modified, whereas a vertical gradient of - 29 W m$^{-2}$ km$^{-1}$ (Marty et al., 2002) was applied to LW↓ irradiances to mitigate the remaining differences in altitude. The different simulations are summarized in Table 2.

### 2.3.2 In situ snowpack observations

To assess the quality of Crocus simulations, an observational dataset of SD measurements was constituted in the French Alps and the Pyrenees, within SAFRAN massifs. Only stations with less than 150 m elevation difference to the model topography were selected. This dataset contains a total of 172 stations (89 in the French Alps and 83 in the Pyrenees) with daily manual measurements at ski resorts (at 6 UTC) and daily automatic measurements by ultra-sonic sensors at high altitude sensors, as described in details in Vionnet et al. (2016) for the French Alps and in Quéno et al. (2016) for the French and Spanish Pyrenees.

## 3 Evaluation of radiation products over the Alps and the Pyrenees

### 3.1 Comparisons with in situ measurements

SW↓ and LW↓ irradiances from LSA SAF products, AROME forecasts and SAFRAN reanalyses were evaluated using in situ measurements. The altitude of the grid points associated to each station is reported in Table 1. Biases and Root Mean Square Errors (RMSE) were computed in absolute and relative values (with the mean of observations as reference). To account for topographic shading, a topographic mask was computed after a 25 m resolution digital elevation model at all stations except Andorre and Envalira. The SW↓ irradiance products were only evaluated when the sun was above the horizon, or when the observed value was higher than 20 W m$^{-2}$ at Andorre and Envalira stations. The LW↓ irradiance products were evaluated by day and night.

The SW↓ scores for all stations are listed in Table 1. For most stations, DSSF shows the lowest biases with an underestimation of SW↓ (until - 15% at Argentière glacier). Biases are also mostly negative for SAFRAN (until - 25% at Villar-St-Pancrace), while AROME exhibits strong positive biases at most of the stations (until + 24% at Col de Porte). DSSF exhibits the lowest RMSE at all stations except Col de Porte and Argentière glacier. For all products, the lowest RMSE are reached at Carpentras in plains. These scores are summarized in Fig. 2. The distinction by domain (French Alps and Pyrenees) shows that the three products have very similar RMSE over both domains, which highlights the consistency of these scores. The distinction by range of altitude (1000 m - 1500 m,





1500 m - 2000 m, > 2000 m) shows increasing RMSE with altitude for DSSF, while RMSE are higher but more constant for AROME and SAFRAN. The increasing RMSE of DSSF is mainly due to stronger negative biases at high altitudes (- 39 W m$^{-2}$ above 2000 m against - 8 W m$^{-2}$ between 1000 m and 1500 m). SAFRAN biases are negative at all altitudes while AROME biases are positive at all altitudes. Overall, DSSF exhibits the best scores with a relative bias of - 4% and a relative RMSE of 33%. SAFRAN has a relative bias of - 7% and a relative RMSE of 40%. Finally, AROME exhibits the strongest relative bias (+ 12%) and the highest relative RMSE (43%).

Fig. 3 shows biases and RMSE of the different datasets of incoming LW↓ (DSLF, DSLFnew, AROME and SAFRAN) at the five LW↓ stations and the overall scores. In this figure, stations are ordered by altitude. In mountains, DSLF, DSLFnew and AROME have a negative bias, while SAFRAN bias tends to increase with altitude (from - 7 W m$^{-2}$ at Col de Porte to + 19 W m$^{-2}$ at St-Sorlin glacier). At low elevation (Carpentras), the best scores are in favour of DSLFnew with a bias of + 4 W m$^{-2}$ (+ 1%) and a RMSE of 16 W m$^{-2}$ (5%), which falls within the range of uncertainties of the sensor. At three mountain stations (Col de Porte, Bassiès and Argentière glacier), the lowest bias and RMSE are reached by SAFRAN, while AROME has the lowest RMSE at St-Sorlin glacier. Overall, AROME exhibits the strongest negative relative bias (- 6%) and the highest relative RMSE (12%). DSLF and DSLFnew have equivalent scores with a relative bias of - 3% and a relative RMSE of 11%. Finally, SAFRAN has a relative bias of + 1% and a relative RMSE of 11%. These global scores are close to the sensor uncertainties in mountains, which does not enable to choose the "best product". However, some trends are identified such as an underestimation of LW↓ by DSLF, DSLFnew and AROME. The performance of LSA SAF products and models is also clearly better in terms of LW↓ than SW↓, because of lower biases and RMSE.

The yearly cycles of SW↓ irradiances are illustrated at Carpentras for reference (Fig. 4a) and at Péone mountain station (Fig. 4b). They show higher RMSE in Spring and Summer for each dataset, lowest RMSE for DSSF and highest RMSE for AROME during the whole year, except in December and January where the three products have equivalent RMSE. This trend was found similar at all stations. No specific trend was observed for the bias. The SW↓ daily cycles (Fig. 4c for Carpentras and Fig. 4d for Péone) show a lower RMSE for DSSF in the middle of the day. SAFRAN cycle is not marked enough (positive biases in the morning and evening, negative biases in the middle of the day). Whatever the hour, AROME overestimates SW↓. DSSF represents well the diurnal cycle, with an underestimation in the afternoon. These trends were also highlighted at the other mountain stations. The study of the daily and yearly cycles of LW↓ irradiances did not indicate any particular trend for scores following the month or the hour (not shown).

### 3.2 Spatial comparisons of the distributed products

Spatial comparisons of the different irradiance products were carried out over the two domains. DSSF and DSLF were taken as references. The spatial distributions of their annual mean computed



using data from 1 August 2010 to 31 July 2014 and the differences with the other irradiance products are shown in Fig. 5 for the French Alps and in Fig. 6 for the Pyrenees.

The DSLF exhibits a strong correlation with the altitude, with a decreasing LW irradiance towards
the highest elevations, i.e. the East of the French Alps (Fig. 5a) and the central range of the Pyrenees (Fig. 6a). AROME presents a moderate negative bias as compared to the DSLF, both in the Alps (Fig. 5b) and in the Pyrenees (Fig. 6b), while SAFRAN presents a strong positive bias, particularly in the highest areas of the Alps (Fig. 5c) and the Pyrenees (Fig. 6c). DLSFnew presents a slight positive bias over most of the domains, except over the highest peaks where the bias is slightly
negative (Fig. 5d and Fig. 6d).

The DSSF exhibits a lower correlation with the topography (Fig. 5e and Fig. 6e). For given sky conditions, the SW irradiance increases with the elevation as the atmospheric transmissivity increases. But the annual mean of the DSSF follows more regional patterns of cloud cover than elevation patterns. For example, in the French Alps, Fig. 5e shows a North-West – South-East gradient
of increasing DSSF: South-Eastern massifs are often shielded by North-Western massifs in the most frequent case of West and North-West disturbed flows. A similar gradient of precipitation was shown in Durand et al. (2009b). The heterogeneity of DSSF is even more marked in the Pyrenees (Fig. 6e) where the West-East chain acts as an orographic barrier to the prevailing westerlies and northwesterlies coming from the Atlantic Ocean (Quéno et al., 2016). A clear discontinuity appears between
the French Pyrenees, where the clouds are often blocked, and the Spanish Pyrenees, often affected by Foehn wind and resulting clear sky conditions. The lowest DSSF are found in the Western part of the French Pyrenees, while the Eastern part is more sunny due to the abating Atlantic influence and a Mediterranean climate. AROME presents a strong positive bias (Fig. 5f and Fig. 6f), locally higher than 30% over the highest peaks, and still higher than 15% in many plain areas. SAFRAN bias is
very variable from one massif to another (Fig. 5g and Fig. 6g). A strong negative bias for SAFRAN can be noticed in the South-Western massifs of the Spanish Pyrenees (Fig. 6g), highlighting a poor representation of the orographic blocking as already noticed in Quéno et al. (2016).

The dependence of the different irradiance products with the altitude was further explored with the study of altitudinal gradients. Figure 7 represents the vertical evolution of the LW↓ and SW↓
averaged over the SAFRAN massifs of the French Alps and the Pyrenees by steps of 100 m of elevation over the whole study period, together with the associated standard deviations.

The strong dependency of LW↓ irradiance with altitude is confirmed in Fig. 7a for the French Alps and Fig. 7b for the Pyrenees. As a reference, the altitudinal gradient for annual LW↓ means of -29 W m$^{-2}$ km$^{-1}$ found by Marty et al. (2002) in the Swiss Alps is plotted in dashed line, while
Table 3 lists the mean altitudinal gradient for each dataset in both domains. All datasets present a steady decrease of LW↓ with altitude, and are close to each other below 1200 m approximately. For higher elevations, SAFRAN annual mean value is significantly stronger than AROME, DSLF and DSLFnew, due to a lower vertical gradient (Table 3). We showed in Sect 3.1 that AROME, DSLF





and DSLFnew had a negative bias at the four mountain stations. This effect may come from a too
strong vertical gradient (Table 3). DSLFnew is larger than AROME and DSLF at all altitudes below
2900 m in the French Alps (Fig. 7a) and 2200 m in the Pyrenees (Fig. 7b) approximately. It gets
lower at the highest altitudes due to a stronger vertical gradient. The stronger vertical gradient of
DSLFnew compared to DSLF is the confirmation that the use of forecasts of higher resolution for
the algorithm takes more into account the topography. The excessive vertical gradient may originate
from the cold bias of AROME near-surface temperatures, enhanced with the altitude (Vionnet et al.,
2016), leading to a strong underestimation of the fluxes by DSLFnew at the highest altitudes.

In terms of SW↓ irradiance, Fig. 7c and Fig. 7d highlight that AROME fluxes are significantly
stronger than SAFRAN and DSSF at all altitudes. SAFRAN is marked by an increase of incoming
SW↓ fluxes with altitude, while AROME and DSSF present a more variable evolution, and par-
ticularly a decrease of the fluxes at the highest altitudes in the Alps (Fig. 7c). This decrease may
reflect the more frequent presence of clouds blocked by the highest peaks. Furthermore, these fig-
ures underline a weaker dependency of SW↓ irradiance with altitude than LW↓ irradiance. Indeed,
the standard deviation of LW↓ at a given altitude is small compared to the total variation of the mean
LW↓ with altitude for all products (Fig. 7a and Fig. 7b), whereas they can reach similar values for
SW↓ (Fig. 7c and Fig. 7d). This spatial variability at a given altitude is particularly marked at low-
and mid-altitudes (< 1800 m) in the Pyrenees for AROME and DSSF, reflecting a good represen-
tation of the strong climate heterogeneity between French and Spanish foothills. SAFRAN, which
gives homogeneous analyses per massif, does not account for the spatial variability within the massif
as is the case for AROME and DSSF.

## 4 Impact of the radiation products on snowpack simulations

Snowpack simulations were performed over four winters from 2010 to 2014 to assess the impact of
the different irradiance datasets as radiative forcing. Table 4 summarizes the bias and RMSE for the
three simulations (A-Cro, AL-Cro and AS-Cro) compared at 172 stations of the French Alps and the
Pyrenees over the period. The scores are aggregated by domain and elevation range. As shown by
Vionnet et al. (2016) and Quéno et al. (2016), A-Cro overestimates the snow depth (+ 38 cm in the
French Alps and + 55 cm in the Pyrenees), with marked RMSE (62 cm in the French Alps and 89 cm
in the Pyrenees). The use of DSSF and DSLFnew as radiative forcing (AL-Cro) increases the bias by
+ 5 cm in the French Alps and + 15 cm in the Pyrenees, while the RMSE is increased by + 10 cm in
the French Alps and + 17 cm in the Pyrenees. On the contrary, the use of SAFRAN radiative forcing
(AS-Cro) gives a lower bias (29 cm in the French Alps and 51 cm in the Pyrenees) and RMSE (59
cm in the French Alps and 88 cm in the Pyrenees). The highest biases and RMSE are reached at high
altitude (≥ 2200 m) by AL-Cro, because of the marked underestimation of DSSF and DSLFnew
at these elevations. The use of SAFRAN irradiances (AS-Cro) tends to reduce the biases of A-Cro,





particularly at the lowest elevations where the higher LW↓ increases the melting during the whole
season. Above 1800 m, the RMSE is not reduced by the use of SAFRAN irradiances (except above
2200 m in the Alps), because the higher LW↓ enhances the melting in winter and the lower SW↓
reduces the melting in spring, which increases the dispersion around the annual bias.

Figure 8 provides an example of snow depth evolution at Albeille station in the French Pyrenees
(2195 m, located in Fig. 9) during one year (2010/2011), as observed and simulated in the three
configurations. The behaviour of the models at this station is typical of most of the stations. The
three simulations overestimate the snow depth. AL-Cro presents the strongest positive bias during
the whole season, because of lower values of LW↓ and SW↓. On the contrary, AS-Cro exhibits
a lower overestimation than the other simulations during all the accumulation period (until mid-
March approximately). It can be explained by the values of SAFRAN LW↓ irradiance, which are
higher than the other datasets. In winter, SW↓ radiations are low and the snow albedo is high: their
contribution to the surface energy budget is much lower than in spring. Thus, LW↓ radiations have
a higher relative contribution during the accumulation period. However, during the melting period
(from mid-March to mid-May here), the contribution of SW↓ radiations is the highest, due to higher
extra-terrestrial solar fluxes, longer days and lower snow albedo: because of their higher SW↓, A-Cro
simulations melt faster than AS-Cro, which reduces their bias.

These trends can also be observed when looking at maps of spatially distributed snowpack simula-
tions. Figure 9 represents the SWE (snow water equivalent) simulated by A-Cro taken as a reference
on 1 February 2013 during the accumulation period and on 1 May 2013 during the melting period,
and the differences between AL-Cro, AS-Cro and this reference at the same dates. The differences
with AL-Cro are generally between - 50 mm and + 50 mm on 1 February 2013. AS-Cro exhibits
lower SWE values at this date, due to its higher LW↓ irradiance. However, on 1 May 2013, both sim-
ulations exhibit higher SWE values than A-Cro almost everywhere, with differences mostly higher
than 200 mm, locally reaching 400 mm, due to lower SW↓ irradiances.

The impact of the radiative forcing on SWE simulations was further studied at two grid points in
the French Pyrenees: one at low altitude (point A, 1359 m) and one at high altitude (point B, 2459
m), both located in Fig. 9. Figure 10 represents the simulated SWE and cumulated melting at point
A during the winter season 2010/2011, together with the difference in irradiance with AROME as
reference. The same evolutions at point B are represented in Fig. 11. The relative impact of DSSF
and DSLFnew is represented in dashed lines (simulations $AL_{SW}$-Cro and $AL_{LW}$-Cro, Table 2).
At point A, melting occurs during the winter. Consequently, AS-Cro and $AL_{LW}$-Cro simulations
lead to lower values of SWE than A-Cro, since they both exhibit higher LW↓ than AROME (+ 8
W m$^{-2}$ for DSLFnew and + 9 W m$^{-2}$ for SAFRAN). Thus, on 15 February 2011, the cumulated
melting is more than doubled for AL-Cro (104 mm, and 154 mm for $AL_{LW}$-Cro) compared to A-Cro
(42 mm). The lower SW↓ of DSSF compared to AROME (- 15 W m$^{-2}$) implies very limited SWE
differences with A-Cro in the heart of the winter (same cumulated melting for A-Cro and $AL_{SW}$-Cro





on 15 February 2011). Similarly, the lower SW↓ of SAFRAN (- 3 W m$^{-2}$) cannot compensate the higher LW↓ during the winter. The simulation using both DSSF and DSLFnew radiations (AL-Cro) is intermediate between both curves (AL$_{LW}$-Cro and AL$_{SW}$-Cro). At high altitude (Fig. 11), the melting period starts at the beginning of April. Thus, there are no differences between all simulations

until then, despite strong differences in the radiative forcing. Snow melts slightly more slowly with SAFRAN radiative forcing, the lower SW↓ being counterbalanced by the higher LW↓. A marked difference in the melt timing can be noted for AL-Cro: the lower SW↓ is not counterbalanced by the slightly higher LW↓. The peak SWE is shifted by almost one month compared to A-Cro. Therefore, it leads to marked differences in terms of cumulated melting: on 1 June 2011, the cumulated melting

for A-Cro reaches 1149 mm, i.e. almost the double of AL-Cro (613 mm, and 433 mm for AL$_{SW}$-Cro). The simulation mixing DSSF and DSLFnew radiations (AL-Cro) is very close to the DSSF-only simulation (AL$_{SW}$-Cro). Overall, the effect of DSSF prevails at high altitude leading to a later end of the snow cover, while the effect of DSLFnew prevails at low altitude leading to an earlier end of the snow cover.

## 5 Discussion

### 5.1 Quality of irradiance datasets in mountainous terrain

We presented an overview of the quality of several irradiance datasets through an in-depth assessment of the irradiance fields in mountainous terrain. In terms of SW↓ irradiances, DSSF exhibits best scores in mountains, particularly below 2000 m. Above 2000 m, its RMSE is similar to SAFRAN

and AROME, due to a strong negative bias. AROME presents systematic and large overestimations of SW↓ irradiances, contrarily to SAFRAN tendency to underestimate them. The spatial variations of SW↓ irradiances are better represented in DSSF and AROME than in SAFRAN. In terms of LW↓ irradiances, the obtained errors are comparable and it is difficult to identify the best product. The use of forecasts at higher spatial resolution to compute DSLFnew enhances the topographic dependence,

which limits the underestimation of LW↓ irradiance at low and mid-altitudes found with DSLF, but strengthens the negative bias at high altitude. The resulting altitudinal gradient is probably too strong. It may originate from the cold bias of AROME near-surface temperatures, enhanced with the altitude (Vionnet et al., 2016), which leads to a strong underestimation of the fluxes by DSLFnew at the highest altitudes.

Several studies evaluated LSA SAF irradiance products at hourly time step (when the sun is above the horizon for SW↓) at plain stations. For DSSF, we showed in this study a bias of - 14 W m$^{-2}$ and a RMSE of 117 W m$^{-2}$, while in plains, Geiger et al. (2008b), Ineichen et al. (2009) and Cristóbal and Anderson (2013) reported biases of + 2 W m$^{-2}$, + 5 W m$^{-2}$ and - 5 W m$^{-2}$ respectively, and RMSE of 87 W m$^{-2}$, 103 W m$^{-2}$ and 65 W m$^{-2}$ respectively. The higher RMSE in mountains

may partly be explained by higher mean values. For DSLF, we showed in this study a bias of - 8



W m$^{-2}$ and a RMSE of 32 W m$^{-2}$, while in plains, Trigo et al. (2010) and Ineichen et al. (2009) reported biases of + 3 W m$^{-2}$ and - 11 W m$^{-2}$ respectively, and RMSE of 25 W m$^{-2}$ and 29 W m$^{-2}$ respectively. The scores in mountains are close to the scores in plains, and lie within the range of uncertainty of LW↓ sensors in mountains (Table 1). Thus, the performance of LSA SAF irradiance
products remains satisfactory compared to previous evaluations of these products in plains.

Hinkelman et al. (2015) similarly evaluated the CERES SYN products at mountain stations for 3 hours averages. In terms of SW↓ irradiance, they showed biases between - 13 W m$^{-2}$ and + 51 W m$^{-2}$ and RMSE between 93 W m$^{-2}$ and 162 W m$^{-2}$. In terms of LW↓ irradiance, they showed biases between - 17 W m$^{-2}$ and + 31 W m$^{-2}$ and RMSE between 24 W m$^{-2}$ and 40 W m$^{-2}$. Despite
a coarser spatial resolution, the obtained irradiance errors are similar to those of LSA SAF products, but they are reduced by the 3 hours average. Reaching similar performance at hourly time step can then be considered as an improvement. The shorter time step of LSA SAF products also enables a finer representation of the SW↓ diurnal cycle.

These results suggest that LSA SAF satellite estimates of SW↓ and LW↓ irradiances are suitable to
drive distributed snowpack simulations in mountainous terrain. DSLF can be replaced by DSLFnew up to mid-altitudes (2200 m approximately), where the performance is improved. These products constitute beneficial alternatives to NWP and analysis systems in complex terrain.

### 5.2 Sensitivity of snowpack simulations to the radiative forcing

DSSF and DSLFnew irradiance datasets were used to replace AROME irradiance forecasts as ra-
diative forcing of Crocus simulations. The rest of the atmospheric forcing was taken from AROME forecasts. A similar experiment was done with SAFRAN irradiances. The performance of the snow-pack simulations was degraded when using DSSF and DSLFnew products, with an increased positive snow depth bias. On the contrary, the use of SAFRAN irradiances was found to decrease the positive bias obtained with AROME-Crocus. Vionnet et al. (2016) and Quéno et al. (2016) already showed
an overestimation of snow depth by AROME-Crocus in the French Alps and the Pyrenees respectively. In addition, Quéno et al. (2016) partly attributed this overestimation to an underestimation of strong melting. Thus, replacing AROME radiation forecasts by lower or equivalent values (DSSF and DSLFnew) logically enhances the overestimation, despite the better quality of the new radiation products. In this case, improving the radiation forcing leads to degraded snowpack simulations. This
effect may be attributed to error compensations within the atmospheric forcing and/or within the snowpack model:

– The positive snow depth bias is not due to an overestimation of snow accumulation by AROME-Crocus, as shown by Quéno et al. (2016). The strong overestimation of SW↓ by AROME shown in this study would also tend to increase the melting and reduce the snow depth bias.
We showed here it is not counterbalanced by the underestimation of LW↓. However, the under-estimated melting may be linked to an underestimation of the turbulent fluxes, with a possible





influence of the $T_{2m}$ cold bias, particularly marked at the highest altitudes (- 2.8 K above 2500 m ; Vionnet et al., 2016). Their influence needs to be further explored.

- Within the snowpack model Crocus, Quéno et al. (2016) showed an underestimation of snow
settling, with a direct effect on snow depth bias. The parametrisation of the albedo evolution also needs to be questioned: Lafaysse et al. (2017) underlined a positive bias of Crocus-simulated albedo at Col de Porte (Fig. 1a), which they partly attributed to the parametrisation of albedo decrease in the visible range as a function of the age of the snow layer and the altitude of the site. An overestimation of the albedo indeed decreases the absorption of solar
energy, hence enhancing the positive snow depth bias.

These results endorse the idea that snowpack ensemble simulations are necessary to mitigate error compensations, as recently developed for Crocus with ESCROC (Ensemble System Crocus; Lafaysse et al., 2017).

The sensitivity of Crocus snowpack simulations to the radiative forcing can be interpreted in the
light of several works quantifying the impact of atmospheric forcing errors on snowpack simulations (Raleigh et al., 2015; Lapo et al., 2015b; Sauter and Obleitner, 2015). First, Sauter and Obleitner (2015) studied the influence of uncertainties on atmospheric forcing variables on simulations of glacier mass-balance using Crocus in the Svalbard islands (European Arctic). They identified LW↓ uncertainty as the main source of variance (50%) of the surface energy balance throughout the year.
However, the prevailing effect of LW↓ compared to SW↓ is specific to high latitudes, because of the lack of solar insolation in winter. In our study, we showed that the new LW↓ forcing from DSLFnew (with a positive bias compared to AROME) had a significant impact on the mass budget during the whole winter at low altitudes (Fig. 10), while the impact was more limited at high altitudes (Fig. 11). It can be explained by decreasing LW↓ irradiances with altitude together with increasing
SW↓ irradiances, leading to a more significant impact of SW↓ at high altitudes. It is also due to the earlier snowmelt at low altitudes, which limits the crucial role played by SW↓ in spring.

Furthermore, the differences between the different radiative forcing datasets mainly consist of biases rather than random errors: a typical example is the difference of SW↓ at high altitudes between AROME and DSSF shown in Fig. 11. Their effect is then cumulated during the whole season, rather
than counterbalanced, which increases their impact. It is consistent with the outcomes of Raleigh et al. (2015) who showed that snowpack models are more sensitive to biases than random errors in the forcings. It was particularly highlighted for incoming radiations by Lapo et al. (2015b). Lapo et al. (2015b) additionally showed the smaller SWE impact of SW↓ biases than LW↓ biases, due to the albedo effect. Our example at high altitude (Fig. 11) does not follow this rule because of
marked SW↓ differences in spring, when the albedo decreases. Finally, although the SWE is not impacted by the differences in incoming radiations at high altitude during the accumulation period (Fig. 11), impacts are to be expected in terms of snow surface temperatures, with possible consequences on the snow metamorphism processes. Lapo et al. (2015b) indeed showed more sensitivity



of the snowpack simulations to radiation errors at the coldest sites when evaluated in terms of snow
surface temperature rather than SWE. Future works could thus focus on the impact of the different
incoming radiation datasets on the surface energy budget and the resulting effects on the snowpack
stratigraphy.

## 6  Conclusions

In this paper, we assessed the quality of satellite-derived incoming radiation products (DSSF for solar
irradiance and DSLF for longwave irradiance) in mountainous terrain, by conducting a thorough
inter-comparison study involving kilometric resolution forecasts from the NWP system AROME
and fields from the SAFRAN analysis system. A new satellite-derived product for LW↓ iradiance
(DSLFnew) was developed using the DSLF algorithm fed by AROME forecasts. An evaluation of all
available products was performed against in situ measurements using four years of data in the French
Alps and the Pyrenees. The result analysis showed that DSSF products are best for solar radiations,
despite an underestimation at the highest altitudes, while AROME is associated with a strong positive
bias and SAFRAN with a negative bias. In terms of longwave radiations, contrasted results were
obtained at the mountain stations, all falling within the range of uncertainty of sensors. A systematic
underestimation by AROME, DSLF and DSLFnew was highlighted. The negative bias of DSLF
was reduced by DSLFnew up to mid-altitudes but enhanced at high altitudes due to a too strong
altitudinal gradient. A spatial comparison of the datasets showed that AROME and DSSF better
represent the spatial variability of SW↓ fluxes in mountains by comparison with SAFRAN. These
results are encouraging and highlight the potential benefits of using DSSF, DSLF and DSLFnew
as radiative forcing for snowpack modelling in mountainous terrain. Their relatively good quality
in mountains as compared to lower altitudes also supports the use of these data as climatological
inputs and/or validation datasets for NWP models over complex domains such as mountains, where
incoming radiation measurements are scarce.

An evaluation of distributed snowpack simulations by Crocus driven by AROME and the different
radiation datasets was then conducted in the French Alps and the Pyrenees. We showed that replac-
ing AROME radiations by DSSF and DSLFnew increased the positive bias of snow depth, despite
an overall better performance of these datasets in terms of incoming radiations. Therefore, an im-
proved meteorological forcing does not ensure more accurate snowpack simulations. This is mostly
due to error compensations within the atmospheric forcing and the snowpack model. Complemen-
tary studies are sorely needed to identify the cause of the underestimated melting, which cannot be
attributed to radiative fluxes. They should tackle factors such as the turbulent fluxes simulated by
AROME-Crocus and the albedo parametrisation in Crocus. Ensemble snowpack modelling would
also enable to account for simulation errors (Lafaysse et al., 2017). Apart from the AROME-Crocus



modelling context, in the light of the quality assessment performed in Sect. 3, there is a clear benefit of using LSA SAF satellite products of incoming radiations for snow cover modelling in mountains.

*Author contributions.* FK, VV and LQ designed the study. LQ was responsible for the modelling strategy and the preparation of the manuscript. FK, VV and IDE helped to analyse the results. All authors contributed to the writing of the manuscript.

*Acknowledgements.* DSSF and DSLF were provided by the EUMETSAT Satellite Application Facility on Land Surface Analysis (LSA SAF; Trigo et al., 2011). The authors are grateful to S. Gascoin (CESBIO) for providing
radiation measurements from Bassiès AWS, and to D. Six (IGE) for providing radiation measurements from St-Sorlin glacier and Argentière glacier AWS (data from GLACIOCLIM program, https://glacioclim.osug.fr). We also thank D. Carrer, J.-L. Roujean and C. Meurey (CNRM) for help with the LSA SAF data, and I. Trigo (IPMA) for informations about the DSLF algorithm. CNRM/CEN is part of LabEx OSUG@2020 (ANR10 LABX56).




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





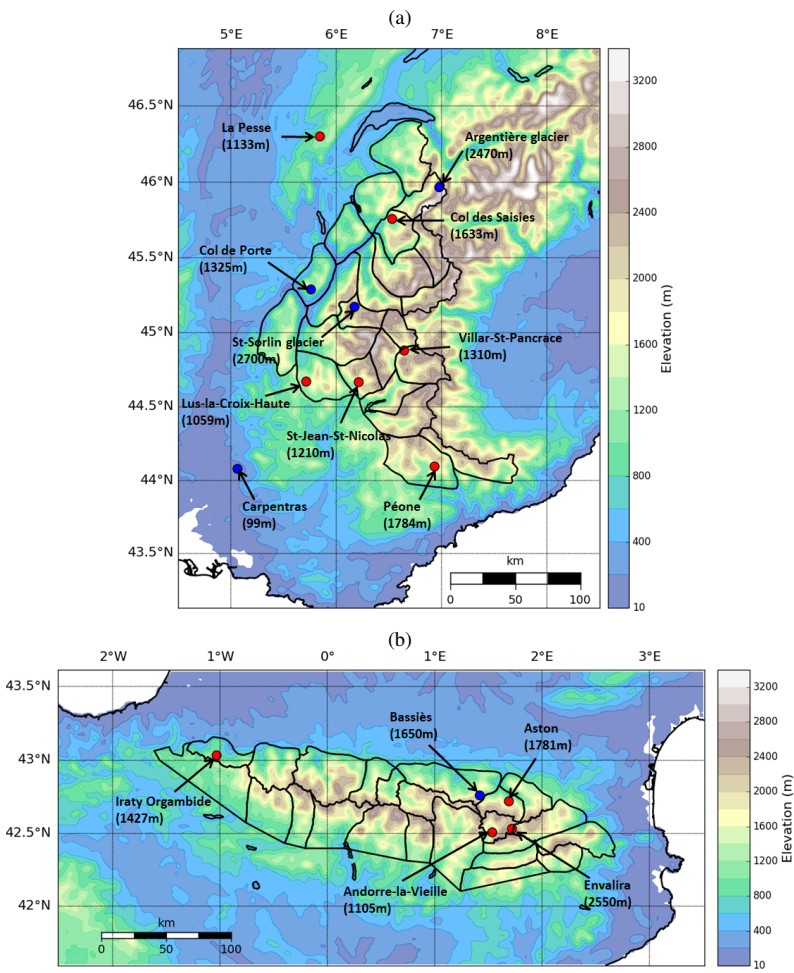

**Figure 1.** Domains of study: (a) the French Alps, (b) the Pyrenees, with AROME topography at 2.5 km resolu-
tion. Red dots: SW↓ stations; blue dots: SW↓ and LW↓ stations; black lines: SAFRAN massifs.




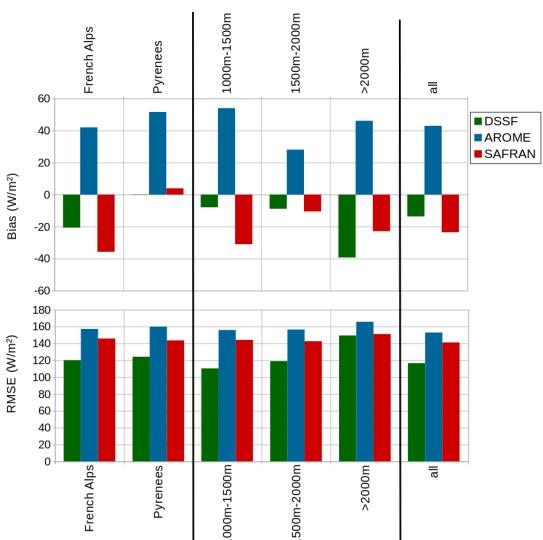

**Figure 2.** Bias and RMSE of SW↓ irradiance products (DSSF in green, AROME in blue, SAFRAN in red) compared to stations gathered by domain (left), range of altitude (center) and all stations (right).

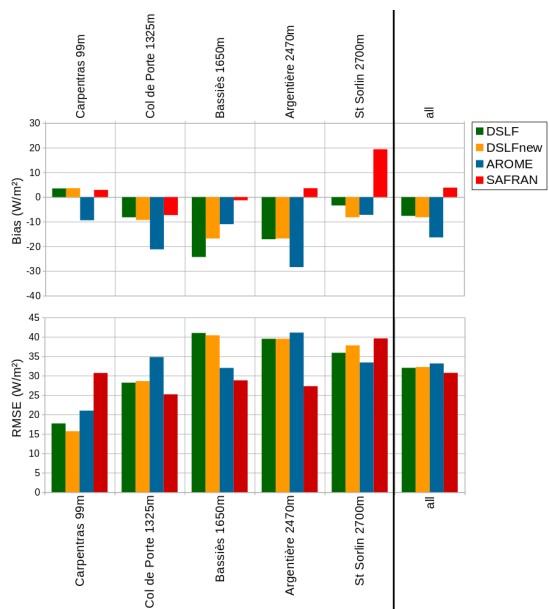

**Figure 3.** Bias and RMSE of LW↓ irradiance products (DSLF in green, DSLFnew in orange, AROME in blue, SAFRAN in red) compared to each station (left) and all stations (right).





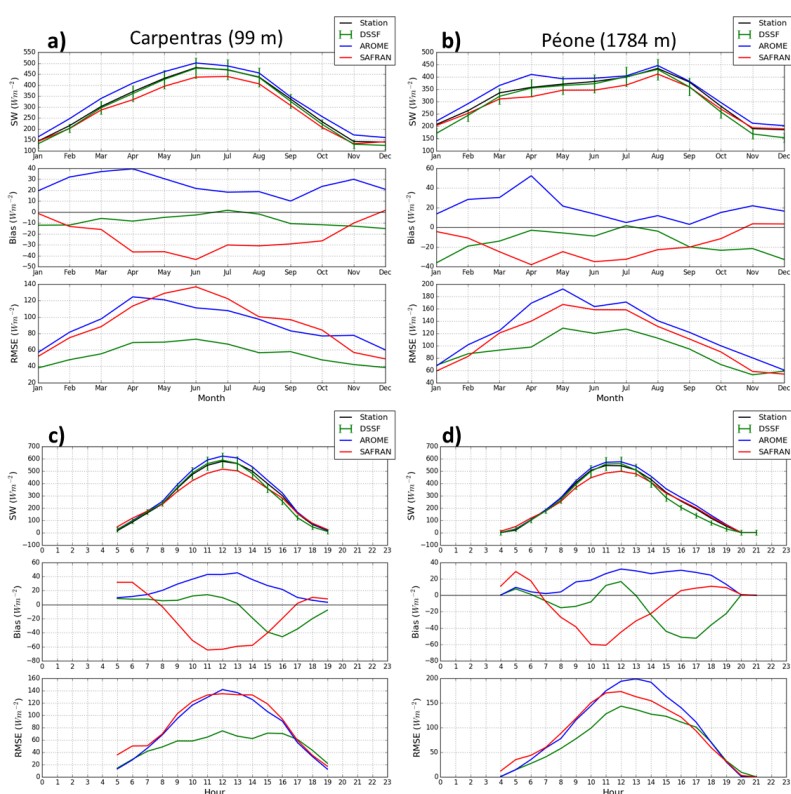

**Figure 4.** Mean yearly cycles of SW↓ irradiance products (DSSF in green, AROME in blue, SAFRAN in red) and ground measurements (in black), bias and RMSE over the 2010-2014 period at: a) Carpentras, b) Péone. Mean daily cycles of the same products, bias and RMSE over the 2010-2014 period at: c) Carpentras, d) Péone.





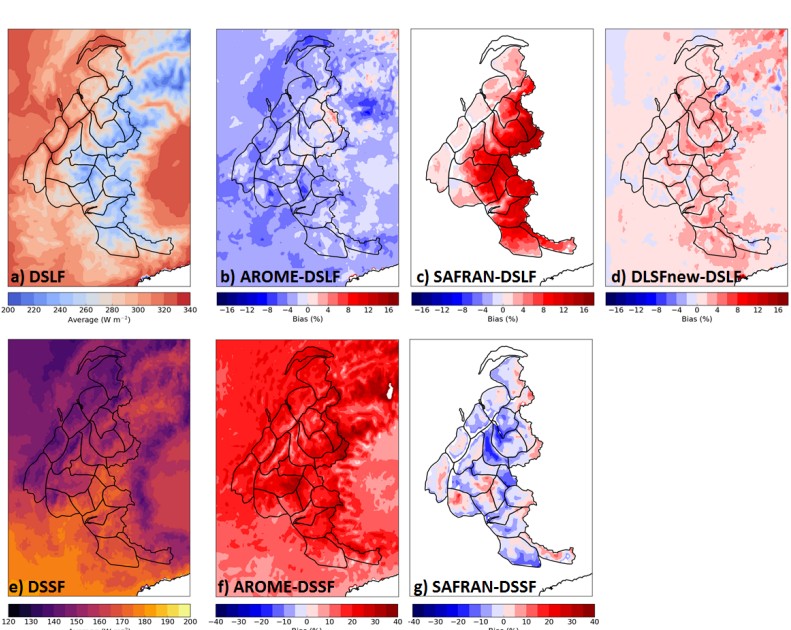

**Figure 5.** a) Average of the DSLF from 1 August 2010 to 31 July 2014 in the French Alps, and relative difference with the DSLF for: b) AROME, c) SAFRAN and d) DSLFnew. e) Average of the DSSF, and relative difference with the DSSF for: f) AROME, g) SAFRAN.



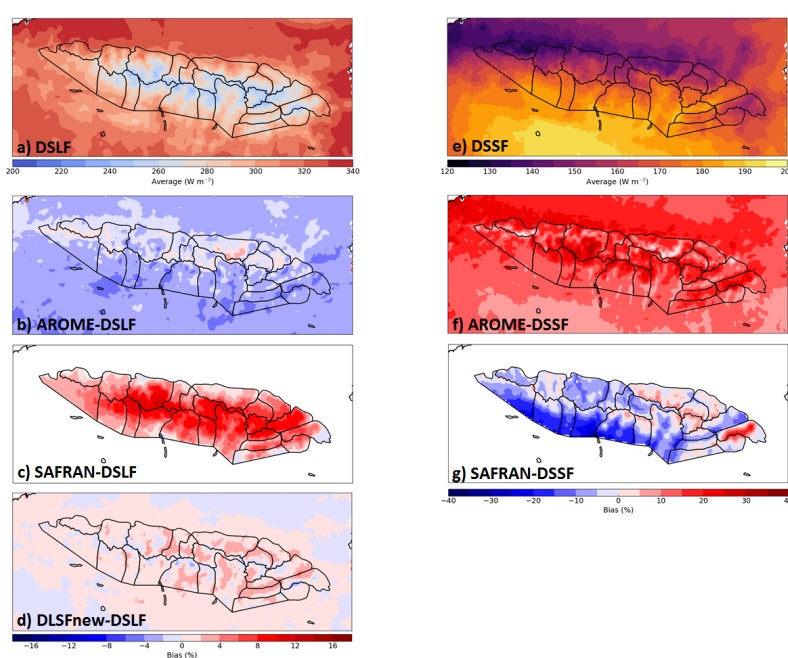

**Figure 6.** a) Average of the DSLF from 1 August 2010 to 31 July 2014 in the Pyrenees, and relative difference with the DSLF for: b) AROME, c) SAFRAN and d) DSLFnew. e) Average of the DSSF, and relative difference with the DSSF for: f) AROME, g) SAFRAN.




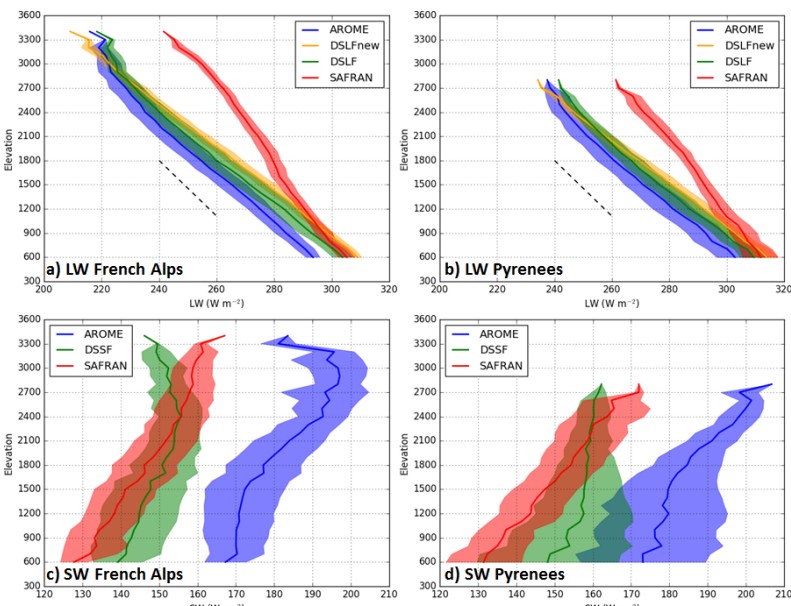

**Figure 7.** Vertical evolution of LW↓ products by steps of 100 m: a) in the French Alps, b) in the Pyrenees, and SW↓ products c) in the Alps, d) in the Pyrenees, averaged over SAFRAN massifs from 1 August 2010 to 31 July 2014, with LSA SAF in green, AROME in blue, SAFRAN in red, DSLFnew in orange. The envelopes represent the mean $\pm$ the standard deviation. The dashed black line represents the climatological LW↓ vertical gradient of -29 W m$^{-2}$ km$^{-1}$ from Marty et al. (2002).

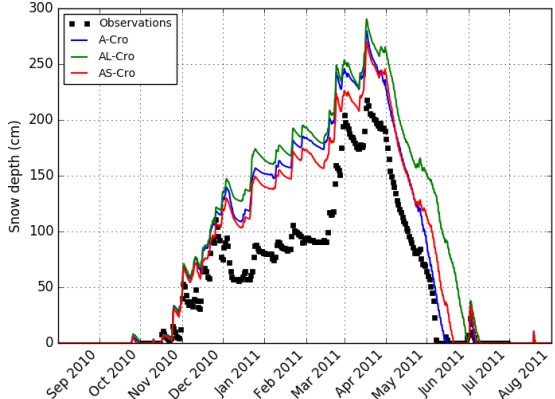

**Figure 8.** Snow depth evolution at Albeille station (2195 m, French Pyrenees) during winter 2010/2011: observations in black, A-Cro simulation in blue, AL-Cro in green, AS-Cro in red.



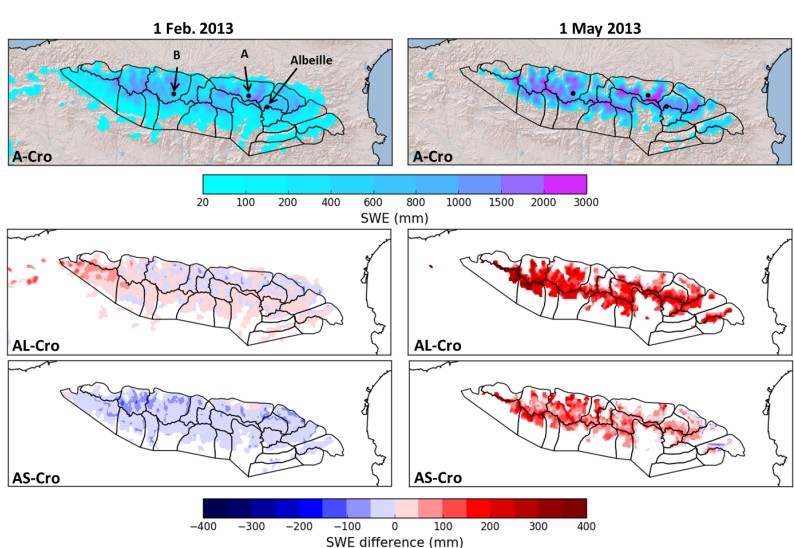

**Figure 9.** Snow Water Equivalent simulated by A-Cro (top) on 1 February 2013 (left) and 1 May 2013 (right) over the Pyrenees. Differences between the SWE simulated by AL-Cro (middle) and AS-Cro (bottom) with A-Cro at the same dates. Points A and B and Albeille station are located.





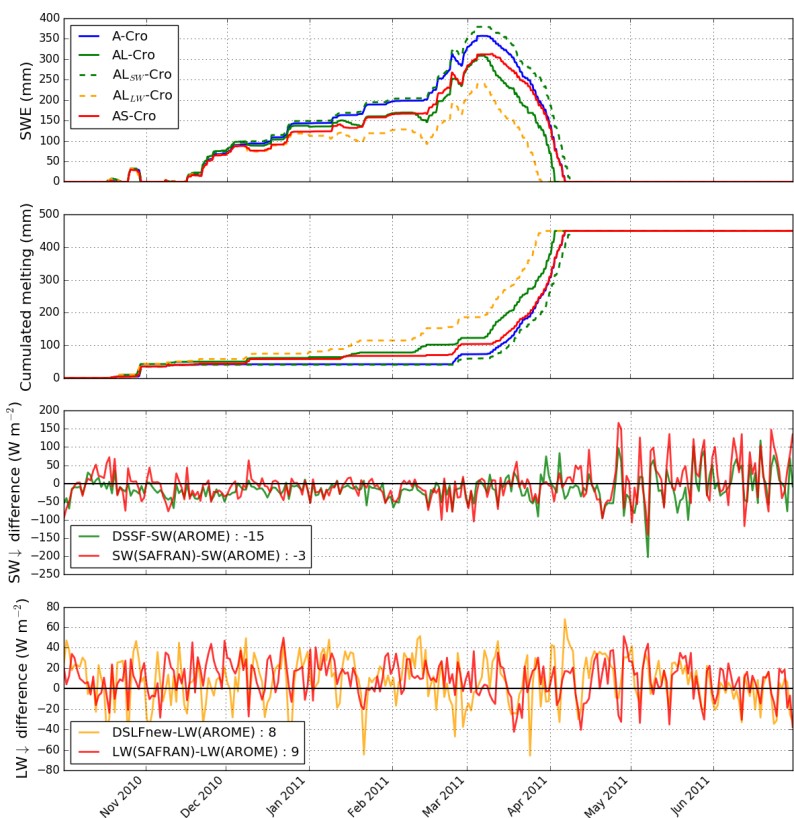

**Figure 10.** Top: Snow Water Equivalent simulated by A-Cro (blue), AL-Cro (green), $AL_{SW}$ (dashed green), $AL_{LW}$ (dashed orange), AS-Cro (red) from 1 October 2010 to 30 June 2011 at point A in the Pyrenees(1359 m, Fig. 9). Middle: Cumulated melting represented with the same colours. Bottom: Mean daily irradiance differences with AROME for DSSF (green), DSLFnew (orange) and SAFRAN irradiances (red).





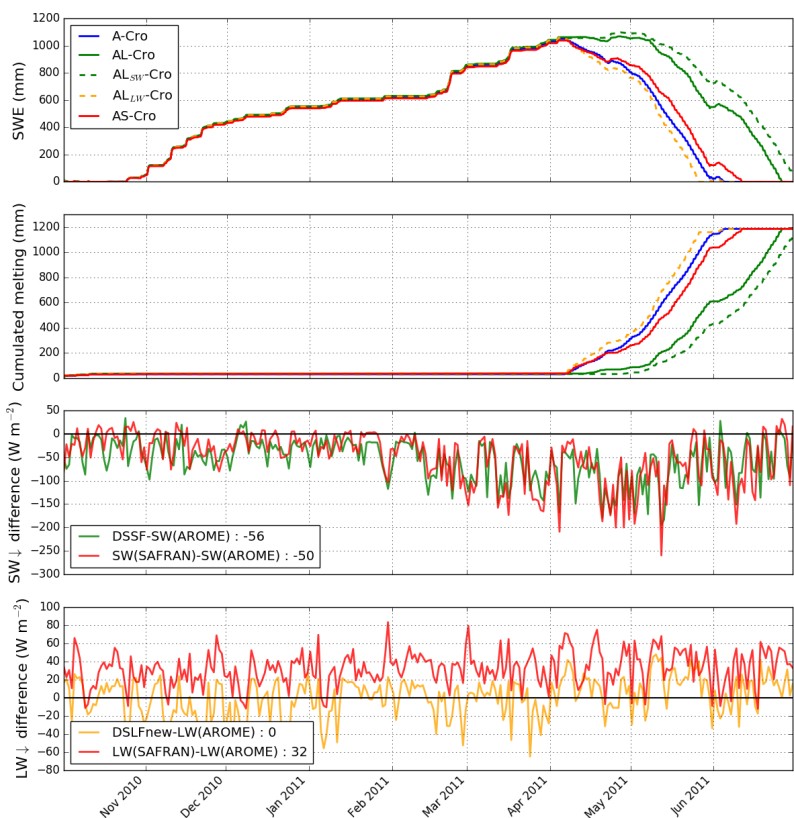

**Figure 11.** Top: Snow Water Equivalent simulated by A-Cro (blue), AL-Cro (green), $AL_{SW}$ (dashed green), $AL_{LW}$ (dashed orange), AS-Cro (red) from 1 October 2010 to 30 June 2011 at point B in the Pyrenees (2459 m, Fig. 9). Middle: Cumulated melting represented with the same colours. Bottom: Mean daily irradiance differences with AROME for DSSF (green), DSLFnew (orange) and SAFRAN irradiances (red).



**Table 1.** List of ground stations, associated mountain range, altitude of the observation, altitude of the associated LSA SAF and AROME grid points, measurement uncertainties, number of hourly SW↓ observations (N), mean observation and bias/RMSE for DSSF, AROME and SAFRAN computed when the sun is not masked, from 1 August 2010 to 31 July 2014. The best scores are given in bold. The mountain range of each station is indicated by A (Alps), J (Jura) or P (Pyrenees).

| station (mountain range) | uncertainties SW↓ | LW↓ | altitude obs. | LSA SAF | ARO. | N | mean SW↓ (W m$^{-2}$) | SW↓ bias (W m$^{-2}$) DSSF | ARO. | SAFR. | SW↓ RMSE (W m$^{-2}$) DSSF | ARO. | SAFR. |
|---|---|---|---|---|---|---|---|---|---|---|---|---|---|
| Carpentras (plains) | ±3% | ±5% | 99 m | 88 m | 99 m | 18 239 | 322 | **-8 (-2%)** | 25 (8%) | -24 (-7%) | **58 (18%)** | 96 (30%) | 99 (31%) |
| Lus-la-Croix-Haute (A) | ±10% | X | 1059 m | 1040 m | 1081 m | 13 616 | 360 | 8 (2%) | 78 (22%) | **-6 (-2%)** | **116 (32%)** | 174 (48%) | 140 (39%) |
| Andorre (P) | ±10% | X | 1105 m | 1073 m | 1385 m | 6 020 | 378 | **-8 (-2%)** | 79 (21%) | -42 (-11%) | **117 (31%)** | 169 (45%) | 155 (41%) |
| La Pesse (J) | ±10% | X | 1133 m | 1131 m | 1119 m | 15 576 | 297 | -18 (-6%) | 38 (13%) | **3 (1%)** | **85 (29%)** | 129 (44%) | 130 (44%) |
| St-Jean-St-Nicolas (A) | ±10% | X | 1210 m | 1197 m | 1315 m | 13 293 | 408 | **-13 (-3%)** | 36 (9%) | -61 (-15%) | **99 (24%)** | 140 (34%) | 146 (36%) |
| Villar-St-Pancrace (A) | ±10% | X | 1310 m | 1412 m | 1521 m | 12 011 | 445 | **-33 (-7%)** | 61 (14%) | -112 (-25%) | **125 (28%)** | 167 (38%) | 191 (43%) |
| Col de Porte (A) | ±10% | ±10% | 1325 m | 1310 m | 1284 m | 7 499 | 392 | 18 (4%) | 96 (24%) | **-8 (-2%)** | 134 (34%) | 202 (52%) | **128 (33%)** |
| Iraty Orgambide (P) | ±10% | X | 1427 m | 1354 m | 1246 m | 16 149 | 273 | **0 (0%)** | 29 (11%) | -7 (-2%) | **110 (40%)** | 136 (50%) | 121 (44%) |
| Col des Saisies (A) | ±10% | X | 1633 m | 1595 m | 1643 m | 11 850 | 355 | -28 (-8%) | **15 (4%)** | -47 (-13%) | **107 (30%)** | 146 (41%) | 132 (37%) |
| Bassiès (P) | ±20% | ±20% | 1650 m | 1785 m | 1714 m | 4 740 | 378 | -17 (-4%) | **-3 (-1%)** | -13 (-3%) | **138 (37%)** | 179 (47%) | 163 (43%) |
| Aston (P) | ±10% | X | 1781 m | 1660 m | 1753 m | 13 859 | 325 | **16 (5%)** | 61 (19%) | 34 (10%) | **140 (43%)** | 176 (54%) | 163 (50%) |
| Péone (A) | ±10% | X | 1784 m | 1754 m | 1704 m | 16 873 | 330 | **-13 (-4%)** | 19 (6%) | -21 (-6%) | **100 (30%)** | 139 (42%) | 125 (38%) |
| Argentière glacier (A) | ±20% | ±20% | 2470 m | 2511 m | 2694 m | 11 565 | 394 | -60 (-15%) | **33 (8%)** | -36 (-9%) | 173 (44%) | 177 (45%) | **165 (42%)** |
| Envalira (P) | ±10% | X | 2550 m | 2577 m | 2394 m | 9 755 | 370 | **-10 (-3%)** | 85 (23%) | 16 (4%) | **120 (32%)** | 157 (42%) | 132 (36%) |
| St-Sorlin glacier (A) | ±20% | ±20% | 2700 m | 2611 m | 2581 m | 10 637 | 430 | -43 (-10%) | **24 (6%)** | -44 (-10%) | **146 (34%)** | 161 (37%) | 153 (35%) |



**Table 2.** Characteristics of the snowpack simulations.

| Simulation names | A-Cro | AS-Cro | AL-Cro | AL$_{SW}$-Cro | AL$_{LW}$-Cro |
|---|---|---|---|---|---|
| Atmospheric forcing (except radiations) | | | AROME | | |
| SW↓ forcing | AROME | SAFRAN | DSSF | DSSF | AROME |
| LW↓ forcing | AROME | SAFRAN | DSLFnew | AROME | DSLFnew |

**Table 3.** Mean altitudinal gradient for AROME, SAFRAN, DSLF and DSLFnew in the French Alps and the Pyrenees.

| | AROME | SAFRAN | DSLF | DSLFnew |
|---|---|---|---|---|
| French Alps | -29 | -21 | -31 | -36 |
| Pyrenees | -31 | -23 | -32 | -37 |

**Table 4.** Bias and root mean square error (RMSE) of snow depth at 172 stations of the French Alps and the Pyrenees over the period 2010-2014 for simulations A-Cro, AL-Cro and AS-Cro. The best scores are given in bold.

| Domain and elevation range | Bias (cm) | | | RMSE (cm) | | |
|---|---|---|---|---|---|---|
| | A-Cro | AL-Cro | AS-Cro | A-Cro | AL-Cro | AS-Cro |
| French Alps | 38 | 43 | **29** | 62 | 72 | **59** |
| < 1800 m | 31 | 29 | **24** | 52 | 53 | **49** |
| [1800 m, 2200 m[ | 26 | 26 | **12** | **58** | 66 | 59 |
| ≥ 2200 m | 61 | 80 | **53** | 79 | 99 | **72** |
| Pyrenees | 55 | 70 | **51** | 89 | 106 | **88** |
| < 1800 m | 66 | 72 | **59** | 97 | 105 | **91** |
| [1800 m, 2200 m[ | 46 | 63 | **43** | **85** | 105 | 86 |
| ≥ 2200 m | 57 | 78 | **56** | **87** | 109 | 89 |