# Peer review of "Satellite-derived products of solar and longwave irradiances used for snowpack modelling in mountainous terrain"

_Hydrology and Earth System Sciences, 2017_

## Referee Comment (RC1) · Anonymous Referee #1 · 14 Nov 2017

Review of "Satellite products of incoming solar and longwave radiations used for snowpack modelling in mountainous terrain"

**General comment**

This manuscript presents an interesting study on evaluating the usage of Meteosat Second Generation satellite derived solar and longwave radiation products in coarse-scale models. For this the radiation products were first compared to ground measurements in the French Alps and the Pyrenees as well as to forecast fields from the AROME model and analyses fields from the SAFRAN system. While the shortwave satellite radiation products showed lower errors with in situ measurements than modelled fields, a clear conclusion for the longwave satellite radiation products could not be drawn (differences around ground measurement uncertainties). Together with forecasts from AROME the satellite radiation products were then used to drive snowpack simulations using the snowpack model Crocus in the French Alps and the Pyrenees. An evaluation with measured snow depth revealed increased biases when using satellite-derived products.

Irradiances derived from the Satellite Application Facility on Land Surface Analysis (LSA SAF) are thoroughly evaluated over mountainous, snow-covered regions, at single points as well as analyzed spatially. The manuscript therefore presents a step towards assimilating high-resolution LSA SAF satellite irradiance products (3 km) over mountainous regions into models with grid cell sizes of only a few kilometers. Overall, the manuscript is well written and I suggest this manuscript to be published once the major comments and corrections listed below were addressed.

**Major comments**

My major comment or concern is about the used satellite-derived products which has an impact on the evaluation applied over mountainous regions. I might be wrong, but I could not find out if the satellite-derived products were corrected for topographical influences on radiation using digital elevation models which is however important.

For instance, spatial de-biasing in the shortwave radiation products needs to be conducted to reduce errors when applied over mountainous terrain. Meteosat Second Generation satellite-derived solar radiation was corrected before, see e.g. the HelioMont method (Stoeckli (2013), Castelli et al. (2014)).

Also, did you consider any topographic corrections for the downward longwave radiation product; I believe the algorithm suggested by Trigo et al. (2010) does not introduce limited sky view.

Please clarify and discuss both.

**Specific comments**

Line 102-104: What are you using the AROME forcing for? Please explain why you are using air temperature and relative humidity in 2m but wind and precipitation in 10m.

Line 120-122: How are the reanalyses interpolated at the exact station locations? Please clarify.

Line 151-152 and Line 160-161 : Please clarify the given target accuracy, i.e. citation. How was it derived ?

Line 170 : If possible, can you add the approximate or range of height of the "first operational atmospheric level" of AROME?

Section 2.2.4 "New DSLF product using AROME forecasts" : What is the reason that you first interpolate the AROME forecasts to the LSA SAF grid? Would it be possible to apply the algorithm directly on the AROME grid assuming the same cloud fraction in all AROME grid cells covered by the coarser LSA SAF grid cell? Maybe this way you could profit from the higher resolution temperature fields as the improvement between DSLF and DSLFnew is not that obvious based on Figure 3.

Line 182-183: I believe the statement that elevation is one of the most significant factors of surface radiation needs clarification. I guess this depends on scale, i.e. represented topographic complexity. There are also differences for shortwave and longwave radiation (as you also found (Figure 7)). Please discuss.

Line 230-233 : Did you evaluate the scenario : shortwave from DSSF and longwave from DSLF ? How do the results compare to those from your scenario c) ?

Line 242-243 : Why did you select a maximum elevation difference of 150 m between AROME grid cell elevation and station elevation for compiling a set of suitable snow depth measurements? In Line 185 you selected a maximum elevation difference of 300 m between AROME grid cell elevation and LSA SAF grid points. What are the reasons for the differing values? Please discuss.

Line 253-255: Please mention briefly which method you used to derive the terrain horizon, e.g. interval size or add a reference.

Line 255: Please specify why the horizon was not computed for Andorre and Envalira. Are those stations without topography in the surroundings?

Line 273: Can you add a similar table for the longwave measurements as Table 1 for the shortwave measurements? The table would give additional inside to the performance with regards to measurement uncertainties and altitude differences between model grid cell and station.

**Technical comments**

Line 40: Consider removing "were".

Figure 4: Please increase all labels and legend.

Line 295: Please rephrase : "Whatever the hour, AROME overestimates SW."

Figure 9: Please rephrase the last sentence in the caption.

Line 442 : Consider refering to Figure 2.

Line 446 : Consider refering to Figure 3.

Line 534-536 : Consider adding to "…due to a too strong altitudinal gradient. " that the gradient arises from the cold bias in AROME air temperatures.

**References :**

Stöckli, R. (2013). The HelioMont surface radiation processing. Scientific report MeteoSwiss, 93, MeteoSwiss: Federal Office of Meteorology and Climatology (123pp.)

Castelli et al. (2014). The HelioMont method for assessing solar irradiance over complex terrain: Validation and improvements. Remote Sensing of Envirnoment 152, 603-613.

---

## Referee Comment (RC2) · Anonymous Referee #2 · 14 Apr 2018

Review of "Satellite products of incoming solar and longwave radiations used for snowpack modelling in mountainous terrain," by Queno, Karbou, Vionnet, and Dombrowski-Etchevers.

Summary: This paper examines radiation data from satellite and model products for their potential use in snowpack modelling. Following the approach of Hinkelman et al., 2015, the general accuracy of the longwave and shortwave irradiances from a forecast model (AROME), a reanalysis product (SAFRAN), and satellite-related data sets (DSSF and DSLF from the LSA SAF) is first assessed through comparisons to measurements at in situ stations in the French Alps and Pyrenees. After this assessment, the various irradiance data sets are used as inputs to a snowpack model (CROCUS) and the accuracy of the respective model runs is evaluated. Based on these analyses, the authors conclude that the most accurate shortwave irradiance values are those in the satellite-related irradiance product while the longwave irradiance products all perform similarly in comparisons to measurements. The snow depth was overestimated in all of the CROCUS runs, with the worst overestimation occurring when the satellite-related data was used as input.

In general, although this study closely follows the lines of Hinkelman et al., 2015, it is useful to show results from different conditions, i.e., different satellite products, different location, different snowpack model, to confirm the results of that study. In addition, detailed evaluation of a number of different irradiance data sets against ground-based measurements and discussion of analyses in terms of altitude are new and useful. I find no major issues with the manuscript and recommend publication after the following comments are addressed. A marked copy of the manuscript with suggested English corrections has been returned to the authors.

Comments:

Lines 141-151. What type of method is used to calculate the shortwave fluxes from the satellite data? I would at least like to know whether it's an explicit radiative transfer calculation, a parameterisation, or something else.

Lines 153-161. Calling the DSLF a satellite product seems a stretch, considering that the underlying algorithm is the Prata parameterisation and the only satellite input is cloud fraction.

Lines 150-151 and 160-161. Can you say whether these targets have been met?

Line 175. What is the source of the $-6.5 \text{ Kkm}^{-1}$ temperature gradient used in computing DSLFnew?

Lines 192-195. The quoted 5-8% accuracy of the Meteo-France pyranometers is probably based on laboratory measurements, not field tests. It would be easier to estimate the actual performance of the Meteo-France instruments if the maintenance regimen was described. It seems unlikely that the uncertainty of these measurements would be the same as those made using better instruments with regular maintenance at Col de Porte.

Line 225. Why were the impact of slope and aspect on the solar irradiance not taken into account in the modeling? Using horizontal irradiances in the comparisons makes sense because all of the data sets provide values in this form, but surely this would have a large impact on the model results. (Incidentally, "supposed to be" suggests that they should meet these conditions, but not that they necessarily do.)

Line 253 says that topographic shading was included in the comparisons to measured irradiance despite the previous statement that slope and aspect could be ignored when running CROCUS. This seems to be a contradiction. Was the shading correction applied to all of the data sets? Was there, in fact, topographic shading at these locations? The method used to make this correction could affect the comparison results.

Lines 267-268. It might be useful to list standard deviations along with biases and RMSEs to allow explicit distinction of the contribution of bias and random error to the RMSE, as was done by Geiger et al., 2008b, Trigo et al., 20ll, and Hinkelman et al., 2015, among others.

Lines 295-296. Based on the shape of the bias plots, it appears that DSSF is out of phase with the measurements. Perhaps you should check the meaning of the time stamps in the satellite data. Misinterpretation could cause the data to be shifted in time relative to the measurements.

Line 318. How can a West-East mountain chain provide a barrier to westerly winds, to create differences on the north and south sides?

Lines 403-404. Please define $AL_{SW}$-Cro and $AL_{LW}$-Cro.

Lines 464-490. The discussion of possible errors in the Crocus model is appreciated.

Lines 491-493. How would using an ensemble of simulations eliminate systematic biases? Do you mean an ensemble of simulations from a number of different models or just one?

Lines 500-501. I don't understand why the greater importance of LW irradiance relative to SW would be "specific to high latitudes." Solar irradiance is also low during the winter in the midlatitudes, so the LW should be of greater importance, at least during the accumulation season. It seems odd that the study by Lapo et al. (2015) is cited in lines 512-514 but then discounted. Although that paper discusses the importance of albedo to the effect of SW irradiance, it also assumes that the changes in the LW and SW energy inputs are similar. Looking at Figure 11, I would not say that the SW is more important than the LW because the albedo is lower in the spring (lines 513-514). Rather, there is a very large SW bias in DSSF (-56 W/m2) and no bias in the DSLFnew LW, such that the total bias is -56 W/m2 relative to AROME. This contrasts with the situation in SAFRAN in which the LW bias offsets that in the SW, yielding a total bias of -18 W/m2.

Lines 553-554. The study did not show that "there is a clear benefit of using LSA SAF satellite products of incoming radiation for snow cover modelling in mountains." To the contrary, the model performed worse when the LSA SFA products were used. Consider changing this to say that, until snowpack models are improved, the LSA SFA products could be used to improve understanding of the models as well as in other snowpack related studies because they provide irradiance data of reasonable quality in mountainous areas (without measurement stations).

Note:

The word "radiation" can be considered either to refer to a process, and hence derived from a verb form (like "differentiation" or "automation"), or a noncountable noun (like "granite" or "wheat.") As such, it is generally not pluralized. Note that it is also not measureable. Like water, only its characteristics can be measured. The relevant SI quantity is irradiance, measured in units of $W/m^2$. It would thus be better in most cases to stick with "irradiance" or the historically used term "(radiative) flux" unless it is being discussed in general (e.g., "Radiation is important to many land surface processes.")

The word "score" usually refers to a tally of points and is thus usually a unitless integer. It isn't really appropriate to refer to RMSEs or means as "scores." As used in this paper, a better word would be "statistics," or possibly "metrics."

---

## Author Comment (AC1) · 29 Sep 2018

Dear Referee,

Thank you for your review and insightful comments.

You mention in your review "a marked copy of the manuscript with suggested English corrections". We did not receive this document, neither in the interactive discussion, nor by mail. The mail indicated for correspondence (louis.queno@meteo.fr) has been deleted before you sent your review. If you sent your document at this address, could you please send it again at the new correspondence address: louis.queno@gmail.com

or through the interactive discussion? Thank you very much in advance.

A point-to-point answer will soon follow.

Best regards, Louis Quéno, on behalf of all co-authors

---

## Author Comment (AC2) · 26 Oct 2018

**Answer to Referee #1**

- **General comment**

**This manuscript presents an interesting study on evaluating the usage of Meteosat Second Generation satellite derived solar and longwave radiation products in coarse-scale models. For this the radiation products were first compared to ground measurements in the French Alps and the Pyrenees as well as to forecast fields from the AROME model and analyses fields from the SAFRAN system. While the shortwave satellite radiation products showed lower errors with in situ measurements than modelled fields, a clear conclusion for the longwave satellite radiation products could not be drawn (differences around ground measurement uncertainties). Together with forecasts from AROME the satellite radiation products were then used to drive snowpack simulations using the snowpack model Crocus in the French Alps and the Pyrenees. An evaluation with measured snow depth revealed increased biases when using satellite-derived products.**
**Irradiances derived from the Satellite Application Facility on Land Surface Analysis (LSA SAF) are thoroughly evaluated over mountainous, snow-covered regions, at single points as well as analyzed spatially. The manuscript therefore presents a step towards assimilating high-resolution LSA SAF satellite irradiance products (3 km) over mountainous regions into models with grid cell sizes of only a few kilometers. Overall, the manuscript is well written and I suggest this manuscript to be published once the major comments and corrections listed below were addressed.**

We thank the referee for the time dedicated to this review and his insightful comments. We answered below to all his points. His comments are in bold while our answers appear in blue. Changes in the manuscript appear in red.

- **Major comments**

**My major comment or concern is about the used satellite-derived products which has an impact on the evaluation applied over mountainous regions. I might be wrong, but I could not find out if the satellite-derived products were corrected for topographical influences on radiation using digital elevation models which is however important.**
**For instance, spatial de-biasing in the shortwave radiation products needs to be conducted to reduce errors when applied over mountainous terrain. Meteosat Second Generation satellite-derived solar radiation was corrected before, see e.g. the HelioMont method (Stoeckli (2013), Castelli et al. (2014)).**
**Also, did you consider any topographic corrections for the downward longwave radiation product; I believe the algorithm suggested by Trigo et al. (2010) does not introduce limited sky view.**
**Please clarify and discuss both.**

The reviewer is right to underline this concern about topographical influence on radiation in mountains. We will try to answer this concern in three points: 1) limitations of LSA SAF irradiance products in mountains; 2) limitations of the use of LSA SAF irradiance products to provide the radiative forcing for distributed snowpack simulations in mountains; 3) influence of the surrounding topography on the evaluation at stations. These elements are developed in a new section added to the discussion.

1) The Heliomont solar irradiance product (Stöckli, 2013; Castelli et al., 2014) is calculated using the MSG SEVIRI High Resolution Visible (HRV; 0.45-1.1 µm) channel and five other near-infrared and infrared channels (0.6, 0.8, 1.6, 10.8, 12.0 µm). In this method, the satellite data depending on the HRV channel (at 1 km resolution) requires an orthorectification to avoid artificial geometric shifts in terrain due to its high resolution compared to the terrain elevation (Stöckli, 2013), while the satellite data from the other channels (at lower spatial resolution) are not orthorectified. The DSSF and the DSLF only use data from the 0.6, 0.8 and 1.6 µm channels, which do not require orthorectification, similarly to the Heliomont method. These details have been added in the manuscript.

   At a 2.5 km scale, the LSA SAF irradiance products have limitations specific to their use in mountains. Although the satellite observations used (cloud mask, top-of-atmosphere reflectance) have a similar spatial resolution, the products rely on meteorological inputs at a coarser resolution (ECMWF forecasts). To represent the topographical influence on these variables, a downscaling is required. As DSLF highly relies on its meteorological inputs, we used finer resolution meteorological inputs for DSLFnew. DSSF relies more on satellite observations, but still includes in the calculations the total column water vapour (TCWV) forecast by ECMWF. It is the main limitation of DSSF in mountains at kilometric scale, since the atmospheric vapour content depends on the elevation. The next step to improve solar irradiance products would thus be to develop a "DSSFnew" using TCWV forecasts at finer scale. It has been mentioned as a limitation and perspective in the new discussion section.

2) The context of this study is to assess different radiative forcings, including satellite-derived irradiance products, for distributed snowpack simulations at a 2.5 km grid spacing. The local topography (aspect, slope, surrounding terrain) is not taken into account in the snowpack simulations made on a flat terrain, similarly to Vionnet et al. (2016) and Quéno et al. (2016). The aim of these simulations is indeed to represent the mean state of the snowpack over the considered pixel (at a given altitude and a given location in the mountain range), thus discarding subgrid topographical specificities. Indeed, the 2.5 km resolution does not enable to reproduce the distribution of slopes and aspects found in a given mountainous area. Consequently, topographical influences on the incoming radiation such as slope/aspect effects, terrain shadowing, limited sky view factor and terrain reflections for SW↓, terrain thermal radiation and limited sky view factor for LW↓, are not taken into account for these ideal "flat pixel" simulations. Besides, they are not taken into account in the existing radiative forcings (SAFRAN and AROME). The aim of our study is to assess the practical benefits of LSA SAF products to provide a more accurate estimate of solar and longwave irradiances over a 2.5 km wide pixel for snowpack simulations. The main limitation of these simulations is that they do not capture a large part of the irradiance variability, strongly determined by the local topography, because these processes occur at a finer scale, requiring sub-grid radiation parameterisations, as developed by Helbig and Löwe (2012) or in the Heliomont method (Stöckli, 2013; Castelli et al., 2014).

3) The local topography affects in situ measurements used for the evaluation of the products. The station locations are generally set up in flat and open fields, which only partly reduces this influence. An effort was made to mitigate the impact of the surrounding terrain on the observations used for comparison to the irradiance products: for SW↓, the effect of terrain shadowing on direct radiation has been taken into account by discarding periods when the sun was masked by the topography.

However, the effects of limited sky view and reflections on diffuse radiation were too difficult to take into account. The terrain effects on measured LW↓ (limited sky view and terrain thermal radiation) were not computed either. Concerning the influence of different elevations in the comparisons, we preferred to indicate the elevation differences (Table 1) than apply a correction.

--- CHANGES IN MANUSCRIPT (lines 490-531) ---

*5.2 Limitations due to the topographical influence on radiation*

*Limitations to the use of kilometric-resolution irradiance products in complex terrain arise from the high topographical influence on incoming radiation. These limitations are tackled here following three axes: (i) limitations of satellite-derived irradiance products in mountainous terrain, (ii) local topographical effects on radiation in the radiative forcing of snowpack simulations, and (iii) influence of local topography on the evaluation of the irradiance products and snowpack simulations.*

*First, satellite data sometimes require corrections when applied over mountains. For instance, the HelioMont solar irradiance product (Stöckli, 2013; Castelli et al., 2014) is calculated using the MSG SEVIRI High Resolution Visible (HRV; 0.45-1.1 μm) channel and five other near-infrared and infrared channels (0.6, 0.8, 1.6, 10.8, 12.0 μm). In this method, the satellite data depending on the HRV channel (at 1 km resolution) requires an orthorectification to avoid artificial geometric shifts in terrain due to its high resolution compared to the terrain elevation (Stöckli, 2013), while the satellite data from the other channels (at MSG pixel resolution, i.e. more than 3 km) are not orthorectified. The DSSF and the DSLF only use data from the 0.6, 0.8 and 1.6 μm channels, which do not require orthorectification, similarly to the HelioMont method. Corrections may also be applied to the meteorological inputs. The DSSF does not rely as much as the DSLF on meteorological forecasts but it still uses the total column water vapour content (TCWV) forecast from ECMWF IFS at 16 km resolution. Since the TCWV is dependent on the elevation, the DSSF could be improved with AROME forecasts of TCWV at kilometric resolution, similarly to DSLFnew. Despite that, the DSSF still exhibits a better performance than AROME and SAFRAN in mountains.*

*At sub-kilometric scale, the local topography strongly influences the solar and longwave irradiance variability. Oliphant et al. (2003) identified the following surface characteristics as causes of radiative flux variability, by order of importance: slope aspect, slope angle, elevation, albedo, shading, sky view factor, and leaf area index. These local factors are not taken into account in AROME, SAFRAN and LSA SAF irradiance products. This study aims at assessing the practical benefits of different irradiance datasets to be used as radiative forcing for distributed snowpack simulations at 2.5 km resolution in mountains. In the context of representing the mean state of the snowpack over a considered flat pixel, at a given altitude and a given location in the mountain range, the terrain influence on the radiation does not need to be taken into account in the radiative forcing. However, to capture the sub-kilometric variability of the snowpack, it will be necessary to consider sub-grid effects of the surrounding terrain on the radiation, and thus a topographical correction of irradiance products (e.g. Helbig and Löwe, 2012) as done for MSG satellite-derived solar fluxes by the HelioMont method (Stöckli, 2013; Castelli et al., 2014).*

*The main limitation implied by local topography effects regards the evaluation of the irradiance products and the snowpack simulations through in situ comparisons. Indeed, in situ irradiance and snow depth measurements are affected by these effects. The location of stations in flat and open fields reduces the impacts of slope, aspect and vegetation. The evaluation of solar irradiances at periods when the sun is not masked by the surrounding topography enables to discard the terrain shadowing effect on direct solar radiation. However, this effect is not considered for snow depth comparisons. Additionally, the limited sky view and the reflection effects on diffuse solar radiation are not taken into account, as well as the limited sky view and terrain thermal radiation effects on longwave irradiance.*

- **Specific comments**

**Line 102-104: What are you using the AROME forcing for? Please explain why you are using air temperature and relative humidity in 2m but wind and precipitation in 10m.**

The word "forcing" was indeed used with no further explanation. We have explained in the new version of the manuscript the use of the forcing to drive snowpack simulations.

The air temperature and humidity are taken at 2 m above the ground and the wind at 10 m above the ground because it corresponds to the heights of the diagnostic variables provided by AROME. The detailed snowpack model Crocus uses forcings taken at these heights when driven by AROME. The precipitation is taken at ground level. It has been precised in the new version of the manuscript.

--- CHANGES IN MANUSCRIPT (lines 102-108) ---
*In this study, we built a continuous atmospheric forcing dataset to drive snowpack simulations using hourly AROME forecasts issued from the 0 UTC analysis time, from + 6 h to + 29 h, extracted on a regular latitude/longitude grid with a 0.025resolution over the period and domains of study (Sect. 2.1, Fig. 1), similarly to Quéno et al. (2016) and Vionnet et al. (2016). Besides incoming shortwave and longwave irradiances, 2 m temperature and humidity, as well as 10 m wind speed and ground-level precipitation (amount of rainfall and snowfall) are part of the AROME forcing. The variable heights correspond to the heights of the diagnostic variables provided by AROME.*

**Line 120-122: How are the reanalyses interpolated at the exact station locations? Please clarify.**

The reanalyses are interpolated at the station locations through a weighted mean of SAFRAN reanalyses at the two closest elevation levels in the considered massif. It has been mentioned in the new manuscript.

--- CHANGES IN MANUSCRIPT (lines 124-127) ---
*For comparisons to in situ irradiance observations, the reanalyses were interpolated at the exact elevation of the stations, through a weighted mean of SAFRAN reanalyses at the two closest elevation levels in the considered massif.*

**Line 151-152 and Line 160-161 : Please clarify the given target accuracy, i.e. citation. How was it derived ?**

The target accuracy is derived from a quality control of both products through comparisons with radiation measurements of the BSRN (Baseline Surface Radiation Network; Ohmura et al., 1998). A citation of the Product Requirement Document (Trigo and Viterbo, 2009) has been added.

--- CHANGES IN MANUSCRIPT (lines 157-159) ---
*The target accuracy of the DSSF is 10% or 20 W $m^{-2}$ for values lower than 200 W $m^{-2}$ (Trigo and Viterbo, 2009).*

--- CHANGES IN MANUSCRIPT (lines 171-172) ---
*The target accuracy of the DSLF is 10% (Trigo and Viterbo, 2009).*

**Line 170 : If possible, can you add the approximate or range of height of the "first operational atmospheric level" of AROME?**

More exactly, air temperature and dew point are taken at 20 m above ground in the archive of the AROME operational forecast. This height corresponds approximately to the height of the

first prognostic level in the operation version of AROME over the period 2010-2014 (Seity et al., 2011). It has been added in the new manuscript.

--- CHANGES IN MANUSCRIPT (lines 180-183) ---
*Air temperature and dew point were taken at 20 m above ground in the archive of the AROME operational forecast. This height corresponds approximately to the height of the first prognostic level in the operation version of AROME over the period 2010–2014 (Seity et al., 2011).*

**Section 2.2.4 "New DSLF product using AROME forecasts" : What is the reason that you first interpolate the AROME forecasts to the LSA SAF grid? Would it be possible to apply the algorithm directly on the AROME grid assuming the same cloud fraction in all AROME grid cells covered by the coarser LSA SAF grid cell? Maybe this way you could profit from the higher resolution temperature fields as the improvement between DSLF and DSLFnew is not that obvious based on Figure 3.**

We chose to interpolate AROME forecasts to the LSA SAF grid because we wanted to generate a new product (DSLFnew) on the exact same grid as DSLF, in order to enable direct comparisons (e.g. in Fig. 5d and Fig. 6d). This product is thus on the same grid as the observed cloud mask, which provides the most important input to derive the LW irradiance.

--- CHANGES IN MANUSCRIPT (lines 186-188) ---
*The new product was generated on the exact same grid as DSLF, in order to enable direct comparisons, so AROME forecasts were interpolated over the LSA SAF grid through a closest-neighbour method (similar grid spacing).*

**Line 182-183: I believe the statement that elevation is one of the most significant factors of surface radiation needs clarification. I guess this depends on scale, i.e. represented topographic complexity. There are also differences for shortwave and longwave radiation (as you also found (Figure 7)). Please discuss.**

We agree with the reviewer that stating "elevation is one of the most significant factor of surface radiation variability" would require an additional discussion. However, this discussion would have no place in this section of dataset description. So the sentence was reformulated and a discussion about factors of surface radiation variability is made in the new discussion paragraph.

--- CHANGES IN MANUSCRIPT (lines 196-198) ---
*As elevation influences incoming radiation (Oliphant et al., 2003), stations were not used for evaluation if the difference between the station elevation and the elevation of the four closest AROME and LSA SAF grid points was higher than 300 m.*

**Line 230-233 : Did you evaluate the scenario : shortwave from DSSF and longwave from DSLF ? How do the results compare to those from your scenario c) ?**

This scenario was not evaluated, because of the similarity of DSLF and DSLFnew, compared to the other LW irradiance products. DSLFnew was chosen because of it mitigates the negative bias of DSLF up to 2200 m.

**Line 242-243 : Why did you select a maximum elevation difference of 150 m between AROME grid cell elevation and station elevation for compiling a set of suitable snow depth measurements? In Line 185 you selected a maximum elevation difference of 300 m**

**between AROME grid cell elevation and LSA SAF grid points. What are the reasons for the differing values? Please discuss.**

For snow depth measurements, a maximum elevation difference of 150 m between model grid point and station was chosen in order to keep the same observation dataset as in Quéno et al. (2016) for the Pyrenees and Vionnet et al. (2016) for the Alps. This value enabled to mitigate the differences of snow depth between simulations and observations arising from elevation differences, and keep at the same time a significant and representative ensemble of stations (172).

For irradiance measurements, the tolerance of elevation difference had to be higher due to the scarcity of stations in the Alps and the Pyrenees, in order to keep a representative dataset. As elevation differences up to 300 m can influence the comparisons of irradiances, the altitude of grid points and stations is indicated in Table 1.

A short discussion has been added in the manuscript.

--- CHANGES IN MANUSCRIPT (lines 199-201) ---
*As elevation differences up to 300 m may have an influence on the comparisons, the altitudes of the grid points associated with each station are listed in Table 1 and should be kept in mind when analyzing the evaluation statistics.*

--- CHANGES IN MANUSCRIPT (lines 261-263) ---
*Only stations with less than 150 m elevation difference to the model topography were selected, in order to use the same dataset as Quéno et al. (2016) and Vionnet et al. (2016).*

**Line 253-255: Please mention briefly which method you used to derive the terrain horizon, e.g. interval size or add a reference.**

The terrain horizon is calculated at an interval size of 5°, from a 25 m resolution DEM. Figure R1 provides an example of the terrain horizon at Bassies station.

[Figure]

*Figure R1. Terrain horizon (in degrees) at Bassies station (1650 m, Pyrenees), calculated at 5° intervals.*

--- CHANGES IN MANUSCRIPT (lines 273-277) ---

*To account for topographic shading on irradiance in situ measurements, a topographic mask was computed with a 5° interval size after a 25 m resolution digital elevation model (DEM) of IGN (French National Institute of Geographical and Forest Information), and applied to the SW↓ irradiance products at all stations except Andorre and Envalira, because the DEM of IGN was only available on the French territory.*

**Line 255: Please specify why the horizon was not computed for Andorre and Envalira. Are those stations without topography in the surroundings?**

The horizon was not computed for Andorre and Envalira because the DEM of IGN (French National Institute of Geographical and Forest Information) was not available for these stations located outside of the French territory. These stations are also located in mountains, that is why a threshold SW value was used to discard periods when the sun was masked by the topography. It has been specified in the new manuscript.

**Line 273: Can you add a similar table for the longwave measurements as Table 1 for the shortwave measurements? The table would give additional inside to the performance with regards to measurement uncertainties and altitude differences between model grid cell and station.**

Table 1 actually lists measurement uncertainties and altitude differences between model grid cell and station for LW (columns 3 to 6). Contrary to Fig. 2 where the SW metrics are aggregated by domain and range of altitude, Fig. 3 shows bias and RMSE at each station measuring incoming LW fluxes. We think it would be redundant to add a table with these metrics.

- **Technical comments**

**Line 40: Consider removing "were".**
Done.

**Figure 4: Please increase all labels and legend.**
Done.

**Line 295: Please rephrase : "Whatever the hour, AROME overestimates SW."**
Done.

**Figure 9: Please rephrase the last sentence in the caption.**
Done.

**Line 442 : Consider refering to Figure 2.**
Done.

**Line 446 : Consider refering to Figure 3.**
Done.

**Line 534-536 : Consider adding to "…due to a too strong altitudinal gradient. " that the gradient arises from the cold bias in AROME air temperatures.**
Done.

**References**

Castelli, M., Stöckli, R., Zardi, D., Tetzlaff, A.,Wagner, J., Belluardo, G., Zebisch, M., and Petitta, M.: The HelioMont method for assessing solar irradiance over complex terrain: Validation and improvements, Remote Sens. Environ., 152, 603–613, doi:10.1016/j.rse.2014.07.018, 2014.

Helbig, N. and Löwe, H.: Shortwave radiation parameterization scheme for subgrid topography, J. Geophys. Res. Atmos., 117, doi:10.1029/2011JD016465, 2012.

Ohmura, A., Gilgen, H., Hegner, H., Müller, G., Wild, M., Dutton, E. G., Forgan, B., Fröhlich, C., Philipona, R., Heimo, A., König-Langlo, G., McArthur, B., Pinker, R., Whitlock, C. H., and Dehne, K.: Baseline Surface Radiation Network (BSRN/WCRP): New Precision Radiometry for Climate Research, Bull. Amer. Meteor. Soc., 79, 2115–2136, doi:10.1175/1520-0477(1998)079<2115:BSRNBW>2.0.CO;2, 1998.

Quéno, L., Vionnet, V., Dombrowski-Etchevers, I., Lafaysse, M., Dumont, M., and Karbou, F.: Snowpack modelling in the Pyrenees driven by kilometric-resolution meteorological forecasts, The Cryosphere, 10, 1571–1589, doi:10.5194/tc-10-1571-2016, 2016.

Seity, Y., Brousseau, P., Malardel, S., Hello, G., Bénard, P., Bouttier, F., Lac, C., and Masson, V.: The AROME-France convective scale operational model, Mon. Weather Rev., 129, 976–991, doi:10.1175/2010MWR3425.1, 2011.

Stöckli, R.: The HelioMont Surface Solar Radiation Processing, Tech. Rep. 93, MeteoSwiss, https://www.meteoswiss.admin.ch/content/dam/meteoswiss/de/service-und-publikationen/Publikationen/doc/sr93stoeckli.pdf, 2013.

Trigo, I. and Viterbo, P.: Product Requirement Document, Tech. Rep. 1.11, The EUMETSAT Satellite Application Facility on Land Surface Analysis (LSA SAF), https://landsaf.ipma.pt/GetDocument.do?id=281, 2009.

Vionnet, V., Dombrowski-Etchevers, I., Lafaysse, M., Quéno, L., Seity, Y., and Bazile, E.: Numerical weather forecasts at kilometer scale in the French Alps: evaluation and applications for snowpack modelling, J. Hydrometeor., 17, 2591–2614, doi:10.1175/JHM-D-15-0241.1, 2016.

---

## Author Comment (AC3) · 26 Oct 2018

**Answer to Referee #2**

- **Summary**

**This paper examines radiation data from satellite and model products for their potential use in snowpack modelling. Following the approach of Hinkelman et al., 2015, the general accuracy of the longwave and shortwave irradiances from a forecast model (AROME), a reanalysis product (SAFRAN), and satellite-related data sets (DSSF and DSLF from the LSA SAF) is first assessed through comparisons to measurements at in situ stations in the French Alps and Pyrenees. After this assessment, the various irradiance data sets are used as inputs to a snowpack model (CROCUS) and the accuracy of the respective model runs is evaluated. Based on these analyses, the authors conclude that the most accurate shortwave irradiance values are those in the satellite-related irradiance product while the longwave irradiance products all perform similarly in comparisons to measurements. The snow depth was overestimated in all of the CROCUS runs, with the worst overestimation occurring when the satelliterelated data was used as input.
In general, although this study closely follows the lines of Hinkelman et al., 2015, it is useful to show results from different conditions, i.e., different satellite products, different location, different snowpack model, to confirm the results of that study. In addition, detailed evaluation of a number of different irradiance data sets against ground-based measurements and discussion of analyses in terms of altitude are new and useful. I find no major issues with the manuscript and recommend publication after the following comments are addressed. A marked copy of the manuscript with suggested English corrections has been returned to the authors.**

We thank the referee for the time dedicated to this review and his insightful comments. We answered below to all his points. His comments are in bold while our answers appear in blue. Changes in the manuscript appear in red. We are also grateful to the referee for the suggestions of English corrections; unfortunately, we have not received the marked copy of the manuscript (see previous comment in the interactive discussion).

- **Comments**

**Lines 141-151. What type of method is used to calculate the shortwave fluxes from the satellite data? I would at least like to know whether it's an explicit radiative transfer calculation, a parameterisation, or something else.**

According to the Product User Manual of DSSF (available at: https://landsaf.ipma.pt/GetDocument.do?id=449) and Geiger et al. (2008b), the shortwave fluxes are calculated with a parameterisation of the atmospheric transmittance as a function of the concentration of atmospheric constituents in case of clear sky, and with a simple physical model of radiative transfer using the observed top-of-atmosphere reflectance in case of cloudy sky. We have briefly mentioned these methods in the new manuscript.

--- CHANGES IN MANUSCRIPT (lines 147-155) ---
*Two separate algorithms are then applied. In the clear-sky method, derived from Frouin et al. (1989), the effective transmittance of the atmosphere is* *parameterized* *using the total column water vapour content (TCWV) forecast by the European Centre for Medium-Range Weather Forecasts (ECMWF) Integrated Forecasting System (IFS), the ozone amount from the Total Ozone Mapping Spectrometer climatology, a constant visibility and the surface albedo taken from the LSA SAF albedo product (Geiger et al., 2008a). In the cloudy-sky method, derived from Gautier et al. (1980) and Brisson et al. (1999), the top-of-atmosphere reflectance observed by MSG/SEVIRI is used in addition to the former set of variables* *to apply a simple*

*physical model of radiative transfer.*

**Lines 153-161. Calling the DSLF a satellite product seems a stretch, considering that the underlying algorithm is the Prata parameterisation and the only satellite input is cloud fraction.**

We fully agree with reviewer 2: meteorological variables taken from ECMWF forecasts have a significant weight in the calculation of the DSLF. A sentence recognizing it has been added in the description. However, for the sake of brevity, DSSF and DSLF are called "satellite-derived products" in the rest of the manuscript, including the title.

--- CHANGES IN MANUSCRIPT (lines 168-170) ---
*The DSLF can therefore be described more accurately as a longwave irradiance parameterisation using satellite observations of the cloud mask rather than a satellite product.*

**Lines 150-151 and 160-161. Can you say whether these targets have been met?**

The target accuracy of 10% is not reached most of the time, which is not surprising since it was derived from comparisons at reference plain stations (Trigo and Viterbo, 2009). The performance of satellite-derived products remains satisfactory, compared to AROME and SAFRAN. It has been mentioned in the discussion.

--- CHANGES IN MANUSCRIPT (lines 473-476) ---
*Thus, the performance of LSA SAF irradiance products remains satisfactory compared to previous evaluations of these products in plains, even though they generally do not reach the target accuracy (Sect. 2.2.3), derived from reference plain stations.*

**Line 175. What is the source of the -6.5 Kkm$^{-1}$ temperature gradient used in computing DSLFnew?**

The vertical temperature gradient of -6.5 K km$^{-1}$ comes from the International Standard Atmosphere, now mentioned in the manuscript. We used this value since it is used in the original DSLF product when adjusting ECMWF forecast.

--- CHANGES IN MANUSCRIPT (lines 188-191) ---
*The possible altitude difference between AROME grid points and LSA SAF grid points was mitigated thanks to a vertical temperature gradient of - 6.5 K km$^{-1}$ according to the International Standard Atmosphere, similarly to the method applied to ECMWF IFS forecasts.*

**Lines 192-195. The quoted 5-8% accuracy of the Meteo-France pyranometers is probably based on laboratory measurements, not field tests. It would be easier to estimate the actual performance of the Meteo-France instruments if the maintenance regimen was described. It seems unlikely that the uncertainty of these measurements would be the same as those made using better instruments with regular maintenance at Col de Porte.**

After verification, the 5-8% accuracy of Meteo-France pyranometers is indeed the value based on laboratory measurements. According to the classification of Météo-France stations (Leroy and Leches, 2014), as part of the Radome network, these stations have a required quality of 10% for hourly means. They are classified as category "B" in terms of maintenance, which corresponds to a biennial calibration and a maintenance at least every week if there is staff, every six months otherwise, according to Leroy (2010). In absence of more details concerning each station, we have mentioned in the revised manuscript that the uncertainties may be higher than 10% at these stations.

--- CHANGES IN MANUSCRIPT (lines 208-212) ---

*The pyranometers from Météo-France network (Kipp&Zonen CM5, CM6B and CM11) meet the good quality standards of the World Meteorological Organization (WMO, 2014), hence an uncertainty of hourly total SW↓ irradiance* *of 10% (Leroy and Leches, 2014)*. *Due to their location in altitude,* *the maintenance may not be systematically weekly so that uncertainties of 10% are probably too optimistic*.

**Line 225. Why were the impact of slope and aspect on the solar irradiance not taken into account in the modeling? Using horizontal irradiances in the comparisons makes sense because all of the data sets provide values in this form, but surely this would have a large impact on the model results. (Incidentally, "supposed to be" suggests that they should meet these conditions, but not that they necessarily do.)**

The snowpack simulations are carried out on a 2.5 km grid spacing. This resolution does not enable to reproduce the distribution of slopes and aspects found in a given mountainous area. Therefore, these simulations are made ideally on flat terrain and supposed to provide the mean state of the snowpack over the pixel. Previous distributed snowpack simulations at kilometric scale using Crocus model have been made in this configuration (Vionnet et al., 2016; Quéno et al., 2016). To this end, the meteorological forcing (including the solar and longwave irradiance) has to be provided in the same topographical conditions. In particular, the solar irradiance is provided over a flat terrain, which discards the need to take into account slope and aspect. The influence of local topography on incoming radiation and its impact on snowpack evolution is the scope of finer scale simulations. As this concern was also a part of the other referee's major comment, a new section of the discussion has been added to tackle it.

For comparisons to snow depth measurements, the stations are located on flat terrain: the slope of the concerned pixel could be different if taken into account in the simulation. The main limitation for comparisons to snow depth stations is the local terrain shadowing which cannot be taken into account in the simulation with a 2.5 km grid.

(The expression "supposed to" has been removed, thank you for your remark.)

--- CHANGES IN MANUSCRIPT (lines 510-531) ---

*At sub-kilometric scale, the local topography strongly influences the solar and longwave irradiance variability. Oliphant et al. (2003) identified the following surface characteristics as causes of radiative flux variability, by order of importance: slope aspect, slope angle, elevation, albedo, shading, sky view factor, and leaf area index. These local factors are not taken into account in AROME, SAFRAN and LSA SAF irradiance products. This study aims at assessing the practical benefits of different irradiance datasets to be used as radiative forcing for distributed snowpack simulations at 2.5 km resolution in mountains. In the context of representing the mean state of the snowpack over a considered flat pixel, at a given altitude and a given location in the mountain range, the terrain influence on the radiation does not need to be taken into account in the radiative forcing. However, to capture the sub-kilometric variability of the snowpack, it will be necessary to consider sub-grid effects of the surrounding terrain on the radiation, and thus a topographical correction of irradiance products (e.g. Helbig and Löwe, 2012) as done for MSG satellite-derived solar fluxes by the HelioMont method (Stöckli, 2013; Castelli et al., 2014).*

*The main limitation implied by local topography effects regards the evaluation of the irradiance products and the snowpack simulations through in situ comparisons. Indeed, in situ irradiance and snow depth measurements are affected by these effects. The location of stations in flat and open fields reduces the impacts of slope, aspect and vegetation. The evaluation of solar irradiances at periods when the sun is not masked by the surrounding topography enables to discard the terrain shadowing effect on direct solar radiation. However, this effect is not considered for snow depth comparisons. Additionally, the limited sky view and the reflection effects on diffuse solar radiation are not taken into account, as well as the limited sky view and terrain thermal radiation effects on longwave irradiance.*

**Line 253 says that topographic shading was included in the comparisons to measured irradiance despite the previous statement that slope and aspect could be ignored when running CROCUS. This seems to be a contradiction. Was the shading correction applied to all**

**of the data sets? Was there, in fact, topographic shading at these locations? The method used to make this correction could affect the comparison results.**

The topographic mask is computed at stations to account for the effect of topographic shading on irradiance in situ measurements. It is applied to all the SW irradiance products (DSSF, AROME and SAFRAN) which do not take into account this effect: the comparison with in situ measurements is only made when the sun is above the calculated horizon. This mask only regards the evaluation of SW irradiance products at stations. It has been clarified in this part of the manuscript, and the new discussion section also tackles this topic.

--- CHANGES IN MANUSCRIPT (lines 273-277) ---

*To account for topographic shading on irradiance in situ measurements, a topographic mask was computed with a 5° interval size after a 25 m resolution digital elevation model (DEM) of IGN (French National Institute of Geographical and Forest Information), and applied to the SW↓ irradiance products at all stations except Andorre and Envalira, because the DEM of IGN was only available on the French territory. The SW↓ irradiance products were only evaluated when the sun was above the horizon, or when the observed value was higher than 20 W m$^{-2}$ at Andorre and Envalira stations (to discard periods when the sun is masked by the terrain).*

--- CHANGES IN MANUSCRIPT (lines 510-531) ---

*At sub-kilometric scale, the local topography strongly influences the solar and longwave irradiance variability. Oliphant et al. (2003) identified the following surface characteristics as causes of radiative flux variability, by order of importance: slope aspect, slope angle, elevation, albedo, shading, sky view factor, and leaf area index. These local factors are not taken into account in AROME, SAFRAN and LSA SAF irradiance products. This study aims at assessing the practical benefits of different irradiance datasets to be used as radiative forcing for distributed snowpack simulations at 2.5 km resolution in mountains. In the context of representing the mean state of the snowpack over a considered flat pixel, at a given altitude and a given location in the mountain range, the terrain influence on the radiation does not need to be taken into account in the radiative forcing. However, to capture the sub-kilometric variability of the snowpack, it will be necessary to consider sub-grid effects of the surrounding terrain on the radiation, and thus a topographical correction of irradiance products (e.g. Helbig and Löwe, 2012) as done for MSG satellite-derived solar fluxes by the HelioMont method (Stöckli, 2013; Castelli et al., 2014).*

*The main limitation implied by local topography effects regards the evaluation of the irradiance products and the snowpack simulations through in situ comparisons. Indeed, in situ irradiance and snow depth measurements are affected by these effects. The location of stations in flat and open fields reduces the impacts of slope, aspect and vegetation. The evaluation of solar irradiances at periods when the sun is not masked by the surrounding topography enables to discard the terrain shadowing effect on direct solar radiation. However, this effect is not considered for snow depth comparisons. Additionally, the limited sky view and the reflection effects on diffuse solar radiation are not taken into account, as well as the limited sky view and terrain thermal radiation effects on longwave irradiance.*

**Lines 267-268. It might be useful to list standard deviations along with biases and RMSEs to allow explicit distinction of the contribution of bias and random error to the RMSE, as was done by Geiger et al., 2008b, Trigo et al., 20ll, and Hinkelman et al., 2015, among others.**

As standard deviations of errors can be derived from the values of biases and RMSEs, we have decided not to burden the text, tables and figures with redundant metrics, especially as they do not provide additionnal explanation.

**Lines 295-296. Based on the shape of the bias plots, it appears that DSSF is out of phase with the measurements. Perhaps you should check the meaning of the time stamps in the satellite data. Misinterpretation could cause the data to be shifted in time relative to the measurements.**

We have double-checked the meaning of the time stamps of each dataset and they were not

misinterpreted. DSSF is not really out of phase since the SW daily maximum corresponds to the maximum of the observations. The problem seems to come from an underestimation of SW by DSSF in the afternoon, which we could not explain.

**Line 318. How can a West-East mountain chain provide a barrier to westerly winds, to create differences on the north and south sides?**

The Pyrenees indeed provide a barrier to the northwesterlies and not the westerlies. It has been corrected.

--- CHANGES IN MANUSCRIPT (lines 341-343) ---
*The heterogeneity of DSSF is even more marked in the Pyrenees (Fig. 6e) where the West-East chain acts as an orographic barrier to the prevailing northwesterlies coming from the Atlantic Ocean (Quéno et al., 2016).*

**Lines 403-404. Please define AL$_{SW}$-Cro and AL$_{LW}$-Cro.**

AL$_{SW}$-Cro and AL$_{LW}$-Cro are now defined in Sect. 2.3.1, as well as in Table 2.

--- CHANGES IN MANUSCRIPT (lines 247-252) ---
*The radiative components of the forcings were extracted from the different irradiance datasets: a) AROME irradiance forecasts (simulations named A-Cro hereafter), b) SAFRAN irradiance reanalyses (simulations named AS-Cro hereafter), c) DSSF and DSLFnew (simulations named AL-Cro hereafter), d) DSSF and AROME LW↓ irradiance (simulations named AL$_{SW}$-Cro hereafter), e) DSLFnew and AROME SW↓ irradiance (simulations named AL$_{LW}$-Cro hereafter).*

--- CHANGES IN MANUSCRIPT (lines 427-429) ---
*The relative impact of DSSF and DSLFnew is represented in dashed lines (simulations AL$_{SW}$-Cro and AL$_{LW}$-Cro, as defined in Table 2).*

**Lines 464-490. The discussion of possible errors in the Crocus model is appreciated.**

Thank you.

**Lines 491-493. How would using an ensemble of simulations eliminate systematic biases? Do you mean an ensemble of simulations from a number of different models or just one?**

ESCROC is the multiphysical ensemble system of the snowpack model Crocus (Lafaysse et al., 2017). The ensemble of simulations is thus based on different physical laws for each process within the snowpack, which could eliminate systematic biases. For instance, ESCROC uses several laws for solar radiation absorption and albedo: computing it in three spectral bands (Brun et al., 1992) or using the radiative transfer scheme TARTES (Two-streAm Radiative TransfEr in Snow, Libois, 2014). We have specified in the new manuscript the multiphysical character of this system.

--- CHANGES IN MANUSCRIPT (lines 560-562) ---
*These results endorse the idea that snowpack ensemble simulations are necessary to mitigate error compensations, as recently developed for Crocus with the multiphysical ensemble system ESCROC (Ensemble System Crocus; Lafaysse et al., 2017).*

**Lines 500-501. I don't understand why the greater importance of LW irradiance relative to SW would be "specific to high latitudes." Solar irradiance is also low during the winter in the midlatitudes, so the LW should be of greater importance, at least during the accumulation season.**

We thank the referee for this remark, the sentence was indeed badly formulated. It has been reformulated.

--- CHANGES IN MANUSCRIPT (lines 569-570) ---
*However, the prevailing effect of LW↓ compared to SW↓ is* *more marked at* *high latitudes, because of the lack of solar insolation in winter.*

**It seems odd that the study by Lapo et al. (2015) is cited in lines 512-514 but then discounted. Although that paper discusses the importance of albedo to the effect of SW irradiance, it also assumes that the changes in the LW and SW energy inputs are similar. Looking at Figure 11, I would not say that the SW is more important than the LW because the albedo is lower in the spring (lines 513-514). Rather, there is a very large SW bias in DSSF (-56 W/m2) and no bias in the DSLFnew LW, such that the total bias is -56 W/m2 relative to AROME. This contrasts with the situation in SAFRAN in which the LW bias offsets that in the SW, yielding a total bias of -18 W/m2.**

The reviewer is right to underline this issue. No conclusion can be deduced from Fig. 11 about the albedo effect on the SWE impact of SW and LW biases, because the energy inputs of both terms are indeed very different. Considering this reasoning was wrong and the fact that the outcomes of Lapo et al. (2015b) could not be properly verified in our case with different SW and LW biases, this part of the discussion has been removed.

**Lines 553-554. The study did not show that "there is a clear benefit of using LSA SAF satellite products of incoming radiation for snow cover modelling in mountains." To the contrary, the model performed worse when the LSA SFA products were used. Consider changing this to say that, until snowpack models are improved, the LSA SFA products could be used to improve understanding of the models as well as in other snowpack related studies because they provide irradiance data of reasonable quality in mountainous areas (without measurement stations).**

The reviewer is right: the last sentence of the conclusion did not reflect the results obtained in the study. It has been modified according to the reviewer's suggestion.

--- CHANGES IN MANUSCRIPT (lines 619-623) ---
*Until such improvements are performed in the AROME-Crocus modelling context, the LSA SAF products of incoming radiative fluxes can be used to improve understanding of snowpack models as well as in other snowpack-related studies, because they provide irradiance data of reasonable quality in mountainous areas.*

- **Note**

**The word "radiation" can be considered either to refer to a process, and hence derived from a verb form (like "differentiation" or "automation"), or a noncountable noun (like "granite" or "wheat.") As such, it is generally not pluralized. Note that it is also not measureable. Like water, only its characteristics can be measured. The relevant SI quantity is irradiance, measured in units of W/m2. It would thus be better in most cases to stick with "irradiance" or the historically used term "(radiative) flux" unless it is being discussed in general (e.g., "Radiation is important to many land surface processes.")**

We thank the reviewer for this comment. The text has been corrected accordingly, including the title.

**The word "score" usually refers to a tally of points and is thus usually a unitless integer. It isn't really appropriate to refer to RMSEs or means as "scores." As used in this paper, a better word would be "statistics," or possibly "metrics."**

Thank you for this comment, which has been taken into account.

**References**

Brun, E., David, P., Sudul, M., and Brunot, G.: A numerical model to simulate snow-cover stratigraphy for operational avalanche forecasting, J. Glaciol., 38, 13–22, doi:10.3189/S0022143000009552 , 1992.

Geiger, B., Meurey, C., Lajas, D., Franchistéguy, L., Carrer, D., and Roujean, J.-L.: Near real-time provision of downwelling shortwave radiation estimates derived from satellite observations, Meteor. Appl., 15, 411–420, doi:10.1002/met.84, 2008b.

Lafaysse, M., Cluzet, B., Dumont, M., Lejeune, Y., Vionnet, V., and Morin, S.: A multiphysical ensemble system of numerical snow modelling, The Cryosphere, 11, 1173–1198, doi:10.5194/tc-11-1173-2017, 2017.

Lapo, K. E., Hinkelman, L. M., Raleigh, M. S., and Lundquist, J. D.: Impact of errors in the downwelling irradiances on simulations of snow water equivalent, snow surface temperature, and the snow energy balance, Water Resour. Res., 51, 1649–1670, doi:10.1002/2014WR016259, 2015b.

Leroy, M.: Classification de performance maintenue, Tech. Rep. 37, Météo-France, http://ccrom.meteo.fr/ccrom/IMG/pdf/note_technique37.pdf, 2010.

Leroy, M. and Leches, G.: Classification d'un site, Tech. Rep. 35B, Météo-France, http://ccrom.meteo.fr/ccrom/IMG/pdf/NT035B_V_Nov_2014-3.pdf, 2014.

Libois, Q.: Evolution des propriétés physiques de la neige de surface sur le Plateau Antarctique, Observations et modélisation du transfert radiatif et du métamorphisme, PhD thesis, Université de Grenoble, 2014.

Quéno, L., Vionnet, V., Dombrowski-Etchevers, I., Lafaysse, M., Dumont, M., and Karbou, F.: Snowpack modelling in the Pyrenees driven by kilometric-resolution meteorological forecasts, The Cryosphere, 10, 1571–1589, doi:10.5194/tc-10-1571-2016, 2016.

Trigo, I. and Viterbo, P.: Product Requirement Document, Tech. Rep. 1.11, The EUMETSAT Satellite Application Facility on Land Surface Analysis (LSA SAF), https://landsaf.ipma.pt/GetDocument.do?id=281, 2009.

Vionnet, V., Dombrowski-Etchevers, I., Lafaysse, M., Quéno, L., Seity, Y., and Bazile, E.: Numerical weather forecasts at kilometer scale in the French Alps: evaluation and applications for snowpack modelling, J. Hydrometeor., 17, 2591–2614, doi:10.1175/JHM-D-15-0241.1, 2016.

---

## Referee Report (RR1)

**Review of the Article hess-2017-573**

**Satellite products of incoming solar and longwave radiations used for snowpack modelling in mountainous terrain**

**by Louis Quéno1, Fatima Karbou, Vincent Vionnet, and Ingrid Dombrowski-Etchevers**

This paper compares different products for estimating incoming solar and longwave radiations to be used for snowpack modelling in mountainous terrain using the CROCUS snow model, at the scale of an entire mountain range.

**General comments:**

The accuracy of incoming solar radiation is one of the key source of errors for snowpack modelling. This is particularly true in in mountainous terrain, because complex topography. For this reason, the topic of the paper is relevant. However, this paper seems to be a very technical review of existing products and it adds very little novelty to the science on this topic. I expected indications that are more concrete on how to improve existing products or propose new, better products. Even if the paper is quite boring to read, the analysis is accurate and the discussion is quite interesting. The results part is instead very boring with many technical details, which are not interesting for the general reader. I suggest making this part much more compact, eventually leaving details on the comparison for a supplementary material section. I have some methodological observations on how complex topography has been taken in account in processing the radiation data and on the role of uncertainties in precipitation data (see specific comments).
Nevertheless, I would recommend publication after a careful revision, since the paper could provide useful guidelines on how to interpret remote sensing radiation observations in Alpine regions.

**Specific comments:**

1. **Introduction**

The major source of uncertainty in snow models is likely to be the correct estimation of precipitation and of the rain/snow limit. Then, especially during the melting season, radiation and snow surface radiative properties, and this is the focus of the paper. However, precipitation remains the first uncertainty reason (*Günther et al, 2019; Engel et al., 2017*).

**2. Methods**

In all the compared products, are precipitation and other meteorological variables the same, besides radiation? Otherwise, the comparison could be biased by other sources of errors.

**3. Evaluation of radiation products**

**Line 255.** You justify the fact that the effect of slope and aspect were not taken in account because the evaluation were made over flat terrain. However being in the mountains, sky view factor should be taken

in account. The sky view factor reduces the amount of diffuse shortwave radiation and can affect the longwave radiation balance (i.e. Corripio, 2003; Rigon et al. 2006, eq.5). Was this effect taken in account while evaluating with ground observations?

**3. Impact of products on snowpack simulations**

**Line 363.** Please clarify immediately the meaning of A-Cro, AL-Cro, AS-Cro.

**Line 366.** A-Cro overestimates the snow depth … this can be due different reasons. This should be already clearly stated here. (I´ve seen that later in the discussion this point is addressed).

**5. Discussion**

**Line 477.** Which are the reasons of A-Cro overestimation? Why you state that there is not overestimation of snow accumulation? What about precipitation or snow/rain limits? **Figure 8** seems to suggest that the model´s overestimation is mainly due to an overestimation of the snow precipitation in the accumulation season (or to wind erosion in the snow observations …). In **Figures 8 and 10** it seems also that there are already during the accumulation season differences in the modelled scenarios. Do have all scenarios the same solid precipitation input? Are the differences only caused by the different radiation input?

**Line 482.** Underestimation of the turbulent fluxes can be related to a variety of reasons: surface roughness length, atmospheric stability parametrization, air humidity, temperature and wind biases. Is it possible to discriminate among the different reasons?

**Line 484.** This is a key point for snow modelling. See also the recent work of Günther et al, 2019.

**References**

*Corripio, J.: Vectorial algebra algorithms for calculating terrain parameters from DEMs and solar radiation modelling in mountainous terrain, Int. J. Geogr. Inf. Sci., 17, 1–24, 2003.*

*Engel, M., Notarnicola, C., Endrizzi, S., Bertoldi, G., 2017. Snow model sensitivity analysis to understand spatial and temporal snow dynamics in a high-elevation catchment. Hydrol. Process. 31, 4151–4168. https://doi.org/10.1002/hyp.11314*

*Günther, D., Marke, T., Essery, R., & Strasser, U. ( 2019). Uncertainties in snowpack simulations— Assessing the impact of model structure, parameter choice, and forcing data error on point-scale energy balance snow model performance. Water Resources Research, 55, 2779– 2800. https://doi.org/10.1029/2018WR023403*

*Rigon, R., Bertoldi, G., Over, T.M.T.M.M., 2006. GEOtop: A Distributed Hydrological Model with Coupled Water and Energy Budgets. J. Hydrometeorol. 7, 371–388. https://doi.org/10.1175/JHM497.1*

---

## Author Response (AR2)

**Author's response**

Dear Editor and Referees,

We thank you for providing a detailed review of the revised manuscript. Please find below a point-by-point answer to both reviews, and a marked-up version of the revised manuscript, where changes appear in red.

**Answer to Referee #3**

We thank the referee for his insightful comments. We answered below to all his points. His comments are in bold while our answers appear in blue. Changes in the manuscript appear in red.

**The authors seem to have adequately addressed the comments by the reviewers in the first round. I only have a few minor comments to add:**

**1) What exactly do you mean by 'soil simulations' (p.3, L.72): soil water and temperature, soil chemistry, land surface, ... simulations?**

We here refer to the ISBA land surface model simulations driven by LSA SAF irradiance estimates by Carrer et al. (2012). They showed improvements of the simulated surface and soil temperatures, as well as soil water content, when using LSA SAF products instead of a meteorological analysis product. The sentence has been modified for clarity.

--- CHANGES IN MANUSCRIPT (lines 71-75) ---

*LSA SAF irradiance products were proved to be valuable in plains (e.g. Geiger et al., 2008b; Ineichen et al., 2009; Trigo et al., 2010; Carrer et al., 2012; Moreno et al., 2013; Cristóbal and Anderson, 2013), with a significant positive impact when used for simulations* of the surface and soil temperatures, and soil water content *(Carrer et al., 2012) or evapotranspiration modelling (Ghilain et al., 2011; Sun et al., 2011).*

**2) The DSSF and DSLF products are not corrected for topography effects. Can you indicate which type of bias this is expected to introduce in general?**

When comparing irradiance products to in-situ measurements, the topographical mask was only taken into account to correct direct solar irradiance. As sensors are installed on flat terrain, slope corrections were not relevant. The effect of limited sky view and shortwave radiation reflections on diffuse solar irradiance, and the effect of limited sky view and contribution of the surrounding terrain on longwave irradiance were not taken into account. This limitation was already stated in Sect. 5.2, but we added a short discussion of potential biases, following the reviewer's recommendation.

Following the incoming radiation budget taking into account topography effects (e.g. Müller and Scherer, 2005), the total shortwave irradiance on a flat surface can be written as:

$$SW \downarrow = shading * SW_{dir} \downarrow + SVF * SW_{dif} \downarrow + (1 - SVF) * SW \uparrow$$

with $SW_{dir}\downarrow$ the direct solar irradiance, $SW_{dif}\downarrow$ the diffuse solar irradiance, $SW\uparrow$ the terrain-reflected solar radiation, and SVF the sky view factor.

Hence, the shortwave irradiance with shading correction but no SVF correction ($SW_{noSVF}\downarrow$) is biased following the equation:

$$SW_{noSVF} \downarrow = SW \downarrow + (1 - SVF) * (SW_{dif} \downarrow - SW \uparrow)$$

This bias is small for a SVF close to 1 (radiation stations are installed in locations as open as possible). This bias is small in clear-sky conditions, because the SVF contribution is then significantly lower than topographic shading (e.g. Olson et al., 2019), but it increases in cloudy conditions with a stronger relative contribution of $SW_{dif}\downarrow$. Moreover, the estimation of the surrounding terrain albedo would be prone to high uncertainties as the snow cover is not modelled at a sufficient spatial resolution around the station.

The total longwave irradiance, accounting for topography effects is:

$$LW \downarrow = SVF * LW_{atm} + (1 - SVF) * LW_{terrain}$$

with $LW_{atm}$ the atmospheric longwave irradiance and $LW_{terrain}$ the thermal terrain radiation. Hence, the longwave irradiance with no SVF correction ($LW_{noSVF}\downarrow$) is biased following the equation:

$$LW_{noSVF} \downarrow = LW \downarrow + (1 - SVF) * (LW_{atm} - LW_{terrain})$$

This bias is small for a SVF close to 1. This bias is more significant when $LW_{atm}$ is low (Sicart et al., 2006) but is usually small compared to $LW_{atm}$ on horizontal surfaces (Plüss and Ohmura, 1997), which is the case of the measurement locations. Moreover, the estimation of surface temperature of the surrounding terrain would be prone to high uncertainties as the snow cover is not modelled at a sufficient spatial resolution around the station.

--- CHANGES IN MANUSCRIPT (lines 521-539) ---

*The main limitation implied by local topography effects regards the evaluation of the irradiance products and the snowpack simulations through in situ comparisons. Indeed, in situ irradiance and snow depth measurements are affected by these effects. The location of stations in flat and open fields reduces the impacts of slope, aspect and vegetation. The evaluation of solar irradiances at periods when the sun is not masked by the surrounding topography enables to discard the terrain shadowing effect on direct solar radiation. However, this effect is not considered for snow depth comparisons. Additionally, the limited sky view and the reflection effects on diffuse solar radiation are not taken into account, as well as the limited sky view and terrain thermal radiation effects on longwave irradiance.* *According to the incoming radiation budget accounting for topography effects on a flat surface (e.g. Müller and Scherer, 2005), the bias for not taking into account the sky view factor (SVF) in the shortwave irradiance is (1-SVF)\*(SWdif -SWref ), where SWdif is the diffuse shortwave irradiance and SWref is the terrain-reflected shortwave radiation. This bias is small in clear-sky conditions, because the SVF contribution is then significantly lower than topographic shading (e.g. Olson et al., 2019), but it increases in cloudy conditions with a stronger relative contribution of SWdif. The bias for not taking into account the SVF in the longwave irradiance is (1-SVF)\*(LWatm-LWter), where LWatm is the atmospheric longwave irradiance and LWter is the terrain thermal radiation. This bias is more significant when LWatm is low (Sicart et al., 2006), but is usually small compared to LWatm on horizontal surfaces (Plüss and Ohmura, 1997), which is the case of the measurement locations.*

**3) Snowpack datasets: snow depth measurements are available for comparison. Snowpack simulations: both snow depth and SWE are discussed.**
**Would it not make sense to produce the example figures 10-11 (currently only for SWE) for snow depth at observed locations, rather than at 2 arbitrary locations? This way, we can obtain a consistent picture of what is going on with the snow simulations. Also supplement the results or discussion with how SWE and snow depth are related in the model (i.e. which density assumption is used).**

We first compared the results of different snowpack simulations in terms of snow depth relative to observations (Fig. 8 and Table 4). The impact of the different radiative forcings on the snowpack mass balance was then considered. In this case, snow water equivalent is the relevant variable to consider. We therefore compared the simulations with varying irradiance forcings in terms of SWE, to identify individual impact of each product at two points at different altitudes. With this process, we showed an impact depending on the altitude: differences of SWE as early

as during the accumulation season at low altitude, differences of SWE during the ablation season at high altitude.

--- CHANGES IN MANUSCRIPT (lines 420-424) ---

*The impact of the radiative forcing on SWE simulations was further studied at two grid points in the French Pyrenees: one at low altitude (point A, 1359 m) and one at high altitude (point B, 2459 m), both located in Fig. 9. Contrary to snow depth, comparing SWE simulations enables to study the impact of the different radiative forcing datasets on the snowpack mass balance with no additional uncertainty on snow compaction.*

The compaction scheme of Crocus model is now briefly introduced in the discussion, following the reviewer's suggestion.

--- CHANGES IN MANUSCRIPT (lines 562-565) ---

*Within the snowpack model Crocus, Quéno et al. (2016) showed an underestimation of snow settling, with a direct effect on snow depth bias. The snow compaction scheme used in Crocus depends on the weight of overlying snow, temperature and density of snow, liquid water content and snow grain size (Vionnet et al., 2012).*

**4) Increase the font size on the colorers of fig 5-6**

Done.

**5) Suggestion to improve the scientific presentation: I understand that it is common practice in earlier similar papers to show both the bias and the RMSE, but scientifically, it makes more sense to show the bias and the unbiased RMSE. Right now, the RMSE metrics are dominated by bias.**

We agree with the reviewer that the standard deviation of error (STDE) would give an indication of unbiased RMSE. However, to allow a more practical comparison with previous assessment studies of DSSF and DSLF products in plains (e.g. Trigo et al., 2010; Ineichen et al., 2009; Cristobal and Anderson, 2013), we have chosen to keep the assessments using the RMSE/bias metrics.

**Answer to Referee #4**

We thank the referee for his insightful comments. Unfortunately, given the line number references and the mention of some issues already tackled in the first response to reviewers, it seems that the referee reviewed the initial version of the manuscript instead of the revised version. However, we answered below to all his points, considering that his remarks could be applicable to the revised manuscript to a large extent. His comments are in bold while our answers appear in blue. Changes in the manuscript appear in red.

**This paper compares different products for estimating incoming solar and longwave radiations to be used for snowpack modelling in mountainous terrain using the CROCUS snow model, at the scale of an entire mountain range.**

**General comments:**

**The accuracy of incoming solar radiation is one of the key source of errors for snowpack modelling. This is particularly true in in mountainous terrain, because complex topography. For this reason, the topic of the paper is relevant. However, this paper seems to be a very technical review of existing products and it adds very little novelty to the science on this topic. I expected indications that are more concrete on how to improve existing products or propose new, better products. Even if the paper is quite boring to read, the analysis is accurate and the discussion is quite interesting. The results part is instead very boring with many technical details, which are not interesting for the general reader. I suggest making this part much more compact, eventually leaving details on the comparison for a supplementary material section. I have some methodological observations on how complex topography has been taken in account in processing the radiation data and on the role of uncertainties in precipitation data (see specific comments).**
**Nevertheless, I would recommend publication after a careful revision, since the paper could provide useful guidelines on how to interpret remote sensing radiation observations in Alpine regions.**

About the novelty of the paper, it is to our knowledge the first study using satellite-derived irradiance estimates to drive spatially distributed snowpack simulations over two entire mountain ranges (Pyrenees and French Alps) after the study of Hinkelman et al. (2015). In their study, they used similar products but at a much lower spatial and temporal resolution (~ 110 km and 3 h) than the present study (~ 3 km and 30 min). Secondly, it is also the first assessment of LSA SAF irradiance products over alpine terrain. Thirdly, a new LW irradiance product was generated in this study using AROME forecasts instead of ECMWF forecasts to better fit the spatial variability of incoming LW in mountainous terrain.

We modified the results section in the revised paper to improve its readability, while keeping the presentation of results as exhaustive as possible. The first paragraph of Sect. 3.1 was moved to Sect. 2.2.5, because it described the evaluation methods and not the results. Many details in Sect. 3.1 (e.g. some metrics at given stations) were removed because they were redundant with Table 1 or Fig. 3 which already contain all station metrics. However, we kept some numbers useful to quantify the interpretation of plots (e.g. in Sect. 4).

--- CHANGES IN MANUSCRIPT (lines 228-238, paragraph moved from results to methods) ---

*SW↓ and LW↓ irradiances from LSA SAF products, AROME forecasts and SAFRAN reanalyses were evaluated using these in situ measurements. The altitude of the grid points associated to each station is reported in Table 1. Biases and Root Mean Square Errors (RMSE) were computed in absolute and*

*relative values (with the mean of observations as reference). To account for topographic shading on irradiance in situ measurements, a topographic mask was computed with a 5° interval size after a 25 m resolution digital elevation model (DEM) of IGN (French National Institute of Geographical and Forest Information), and applied to the SW↓ irradiance products at all stations except Andorre and Envalira, because the DEM of IGN was only available on the French territory. The SW↓ irradiance products were only evaluated when the sun was above the horizon, or when the observed value was higher than 20 Wm⁻² at Andorre and Envalira stations (to discard periods when the sun is masked by the terrain). The LW↓ irradiance products were evaluated by day and night.*

--- CHANGES IN MANUSCRIPT (lines 282-447) ---

Several details on metrics redundant with tables and figures removed.

**Specific comments:**

**1. Introduction**

**The major source of uncertainty in snow models is likely to be the correct estimation of precipitation and of the rain/snow limit. Then, especially during the melting season, radiation and snow surface radiative properties, and this is the focus of the paper. However, precipitation remains the first uncertainty reason (Günther et al, 2019; Engel et al., 2017).**

We thank the referee for pointing this out, as we did not explicitly mention other sources of uncertainty in the introduction. Günther et al. (2019) and Raleigh et al. (2015) indeed highlight that precipitation and air temperature are crucial sources of uncertainty in the input data. Additional references are now added in the introduction.

--- CHANGES IN MANUSCRIPT (lines 34-38) ---

*It is crucial to accurately represent them in numerical snowpack simulations, as recent works underlined the strong sensitivity of snowpack simulations to the radiative forcing (Raleigh et al., 2015; Lapo et al., 2015b; Sauter and Obleitner, 2015), together with crucial input variables like precipitation and air temperature (Raleigh et al., 2015; Günther et al., 2019).*

**2. Methods**

**In all the compared products, are precipitation and other meteorological variables the same, besides radiation? Otherwise, the comparison could be biased by other sources of errors.**

With the exception of shortwave and longwave irradiance, all snowpack simulations are driven by the same atmospheric forcing (air temperature, air humidity and wind speed at a given height above the ground, solid and liquid precipitation), coming from short-term AROME forecast. Section 2.3.1 lines 258-270, and Table 2 report on this. Therefore, the unique source of difference between the snowpack simulations is the radiative forcing.

**3. Evaluation of radiation products**

**Line 255. You justify the fact that the effect of slope and aspect were not taken in account because the evaluation were made over flat terrain. However being in the mountains, sky view factor should be taken in account. The sky view factor reduces the amount of diffuse shortwave radiation and can affect the longwave radiation balance (i.e. Corripio, 2003; Rigon et al. 2006, eq.5). Was this effect taken in account while evaluating with ground observations?**

This issue was addressed in our response in the first round of review. To report on potential topographical effects, a new section has been added to the discussion of the revised manuscript (see Sect. 5.2).

Snowpack simulations are made at 2.5 km resolution, on virtually flat grid points. Topographical corrections for snowpack modelling would need to be considered at sub-grid scale (e.g. Helbig and van Herwijnen, 2017).

The main careful steps to be taken concern the comparison of irradiance products to in-situ measurements. Even if the stations are installed in locations as open as possible, their location in mountains implies shading and a limited sky view. The shading has an effect on the direct solar irradiance, and is taken into account in our comparisons through a topographical mask. However, the effect of limited sky view and shortwave radiation reflections on diffuse solar irradiance, and the effect of limited sky view and terrain thermal radiation on longwave irradiance are not taken into account. We stress this point and comment on potential biases in the discussion section.

--- CHANGES IN MANUSCRIPT (lines 521-539) ---

*The main limitation implied by local topography effects regards the evaluation of the irradiance products and the snowpack simulations through in situ comparisons. Indeed, in situ irradiance and snow depth measurements are affected by these effects. The location of stations in flat and open fields reduces the impacts of slope, aspect and vegetation. The evaluation of solar irradiances at periods when the sun is not masked by the surrounding topography enables to discard the terrain shadowing effect on direct solar radiation. However, this effect is not considered for snow depth comparisons. Additionally, the limited sky view and the reflection effects on diffuse solar radiation are not taken into account, as well as the limited sky view and terrain thermal radiation effects on longwave irradiance. According to the incoming radiation budget accounting for topography effects on a flat surface (e.g. Müller and Scherer, 2005), the bias for not taking into account the sky view factor (SVF) in the shortwave irradiance is $(1-SVF)*(SWdif -SWref )$, where SWdif is the diffuse shortwave irradiance and SWref is the terrain-reflected shortwave radiation. This bias is small in clear-sky conditions, because the SVF contribution is then significantly lower than topographic shading (e.g. Olson et al., 2019), but it increases in cloudy conditions with a stronger relative contribution of SWdif. The bias for not taking into account the SVF in the longwave irradiance is $(1-SVF)*(LWatm-LWter)$, where LWatm is the atmospheric longwave irradiance and LWter is the terrain thermal radiation. This bias is more significant when LWatm is low (Sicart et al., 2006), but is usually small compared to LWatm on horizontal surfaces (Plüss and Ohmura, 1997), which is the case of the measurement locations.*

**3. Impact of products on snowpack simulations**

**Line 363. Please clarify immediately the meaning of A-Cro, AL-Cro, AS-Cro.**

The meaning of the snowpack simulation acronyms is already defined in the Sect. 2.3.1, lines 259-264, and in Table 2.

**Line 366. A-Cro overestimates the snow depth … this can be due different reasons. This should be already clearly stated here. (I´ve seen that later in the discussion this point is addressed).**

It is now mentioned in the revised manuscript that several different reasons can cause this overestimation, referring to the discussion section not to repeat the same comments twice.

--- CHANGES IN MANUSCRIPT (lines 387-389) ---

*As shown by Vionnet et al. (2016) and Quéno et al. (2016), A-Cro overestimates the snow depth, with marked RMSE. It is due to several different reasons, further discussed in Sect. 5.3.*

**5. Discussion**

**Line 477. Which are the reasons of A-Cro overestimation? Why you state that there is not overestimation of snow accumulation? What about precipitation or snow/rain limits? Figure 8 seems to suggest that the model´s overestimation is mainly due to an overestimation of the snow precipitation in the accumulation season (or to wind erosion in the snow observations …). In Figures 8 and 10 it seems also that there are already during the accumulation season differences in the modelled scenarios. Do have all scenarios the same solid precipitation input? Are the differences only caused by the different radiation input?**

The statement "The positive snow depth bias is not due to an overestimation of snow accumulation by AROME-Crocus, as shown by Quéno et al. (2016)." refers to one of the outcomes of the cited paper, used here to help interpret the results. It is not a direct result of the present study. In particular, through a categorical study of daily snow depth variations, Quéno et al. (2016) showed that the overestimation of snow depth by AROME-Crocus in the Pyrenees was primarily due to an underestimation of ablation and settling processes. The sentence has been changed to avoid any misinterpretation.

--- CHANGES IN MANUSCRIPT (lines 555-556) ---

*Quéno et al. (2016) showed that the positive snow depth bias is not due to an overestimation of daily snowfall by AROME-Crocus.*

All snowpack simulation scenarios have the same meteorological input except for the shortwave and longwave irradiances, as explained in Sect. 2.3.1. The differences visible during the accumulation season in Fig. 8 and Fig. 10 are only caused by different irradiance input. Some scenarios indeed generate more winter snowmelt due to a different surface energy budget.

**Line 482. Underestimation of the turbulent fluxes can be related to a variety of reasons: surface roughness length, atmospheric stability parametrization, air humidity, temperature and wind biases. Is it possible to discriminate among the different reasons?**

The reviewer is right to point out that there are many possible explanations for underestimating turbulent fluxes, not yet been fully investigated. The text has been reformulated for clarity.

--- CHANGES IN MANUSCRIPT (lines 558-561) ---

*However, the underestimated melting may be linked to an underestimation of the turbulent fluxes. It may have several causes that need to be further explored, e.g. a possible influence of the T2m cold bias, particularly marked at the highest altitudes (- 2.8 K above 2500 m ; Vionnet et al., 2016).*

**Line 484. This is a key point for snow modelling. See also the recent work of Günther et al, 2019.**

The reviewer is right to underline that snow compaction is critical to snowpack modelling. It is particularly true when evaluating snow depth simulations, where errors can arise from either the snowpack mass balance or the settling modelling. For this reason, after comparing simulations to snow depth measurements (Fig. 8 and Table 4), we compared all scenarios together in terms of SWE (Fig. 9, 10, 11). The direct effect of the different irradiance inputs on the snowpack mass balance was thus highlighted.

--- CHANGES IN MANUSCRIPT (lines 420-424) ---

*The impact of the radiative forcing on SWE simulations was further studied at two grid points in the French Pyrenees: one at low altitude (point A, 1359 m) and one at high altitude (point B, 2459 m), both located in Fig. 9. Contrary to snow depth, comparing SWE simulations enables to study the impact of the different radiative forcing datasets on the snowpack mass balance with no additional uncertainty on snow compaction.*

--- CHANGES IN MANUSCRIPT (lines 562-565) ---

*Within the snowpack model Crocus, Quéno et al. (2016) showed an underestimation of snow settling, with a direct effect on snow depth bias. The snow compaction scheme used in Crocus depends on the weight of overlying snow, temperature and density of snow, liquid water content and snow grain size (Vionnet et al., 2012).*

[revised manuscript text omitted]

---

## Author Response (AR3)

**Author's response**

Dear Editor,

We thank you for reviewing our revised manuscript. To meet your request, we added a short paragraph at the end of the introduction to clearly state the three main novel contributions of our paper. Please find hereafter the newly revised manuscript, including changes in red.

Best regards,

On behalf of all coauthors,
Louis Quéno

[revised manuscript text omitted]